# Trabectedin derails transcription-coupled nucleotide excision repair to induce DNA breaks in highly transcribed genes

Kook Son [1,5], Vakil Takhaveev [2,5], Visesato Mor[1], Hobin Yu[1,3], Emma Dillier [2], Nicola Zilio [4], Nikolai J. L. Püllen[2], Dmitri Ivanov [1], Helle D. Ulrich [4], Shana J. Sturla [2] ✉ & Orlando D. Schärer [1,3] ✉

Most genotoxic anticancer agents fail in tumors with intact DNA repair. Therefore, trabectedin, an agent more toxic to cells with active DNA repair, specifically transcription-coupled nucleotide excision repair (TC-NER), provides therapeutic opportunities. To unlock the potential of trabectedin and inform its application in precision oncology, an understanding of the mechanism of the drug's TC-NER-dependent toxicity is needed. Here, we determine that abortive TC-NER of trabectedin-DNA adducts forms persistent single-strand breaks (SSBs) as the adducts block the second of the two sequential NER incisions. We map the 3'-hydroxyl groups of SSBs originating from the first NER incision at trabectedin lesions, recording TC-NER on a genome-wide scale. Trabectedin-induced SSBs primarily occur in transcribed strands of active genes and peak near transcription start sites. Frequent SSBs are also found outside gene bodies, connecting TC-NER to divergent transcription from promoters. This work advances the use of trabectedin for precision oncology and for studying TC-NER and transcription.

Numerous anticancer agents, including cisplatin, exert their therapeutic effects by inducing DNA damage and subsequently inhibiting essential cellular processes such as DNA replication and transcription. These agents do not work well in cancers with intrinsically high DNA repair or upregulated DNA repair as a response to chemotherapy[1,2]. Trabectedin (also called ET743), an antitumor drug used for the treatment of sarcoma and ovarian cancer[3,4], is an unusual case, as it is more potent in certain DNA-repair-proficient cells[5,6]. Derived from the sea squirt *Ecteinascidia turbinata*, trabectedin is a complex natural product known to form adducts with DNA at the $N^2$-position of dG (Fig. 1a)[7–9]. The unique mechanism of trabectedin toxicity stems from the fact that cells with transcription-coupled nucleotide excision repair (TC-NER) defects exhibit resistance to the drug, while TC-NER-proficient cells accumulate DNA breaks following treatment[5,10]. This

suggests that the breaks formed by TC-NER upon trabectedin treatment are more toxic than the original DNA adduct. Although earlier studies on the properties of trabectedin have provided clinically relevant insights for its use, the understanding of the exact mechanisms underlying its toxicity remains limited, restricting the drug's application in precision medicine.

A key to the mechanism of trabectedin toxicity lies in understanding its interaction with NER machinery. NER operates through two sub-pathways: global genome (GG)-NER and TC-NER[11]. GG-NER is initiated by damage sensors XPC-RAD23B and UV-DDB (DDB1-DDB2 complex), which recognize lesions that induce thermodynamic destabilization in a DNA duplex[12]. In contrast, TC-NER is initiated by stalling of an RNA polymerase and driven by CSB, CSA, UVSSA, and ELOF1[13,14]. Though the trabectedin-DNA adduct is bulky and causes a

[1]Center for Genomic Integrity, Institute for Basic Science (IBS), 44919 Ulsan, Republic of Korea. [2]Department of Health Sciences and Technology, ETH Zürich, 8092 Zürich, Switzerland. [3]Department of Biological Sciences, Ulsan National Institute of Science and Technology (UNIST), 44919 Ulsan, Republic of Korea. [4]Institute of Molecular Biology (IMB), 55128 Mainz, Germany. [5]These authors contributed equally: Kook Son, Vakil Takhaveev. ✉e-mail: sturlas@ethz.ch; orlando.scharer@ibs.re.kr

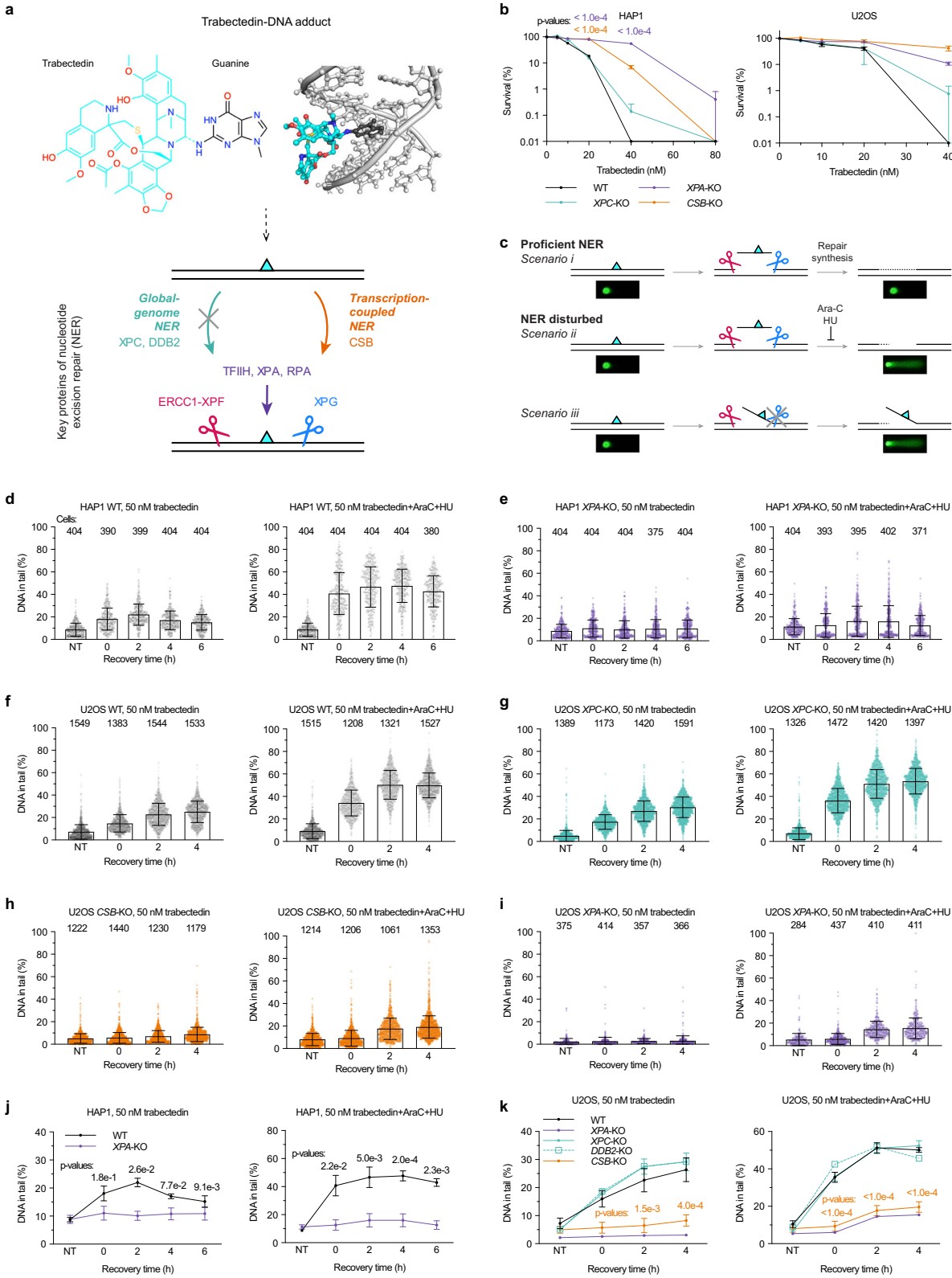

minor bend in the DNA[9,15], it has a thermodynamically stabilizing effect on the DNA duplex[16,17], which is consistent with the lack of recognition by GG-NER[15,18]. Therefore, trabectedin adducts can only be acted upon by the TC-NER pathway (Fig. 1b). A critical unsolved question is how trabectedin induces TC-NER-dependent breaks in DNA, especially given that other TC-NER-specific DNA lesions, for example those formed by illudin S and acylfulvene, undergo complete repair[19–22].

Our goal was to explore the mechanism by which DNA single-strand breaks (SSBs) form and persist following the processing of trabectedin by TC-NER. Utilizing NER-specific alkaline COMET chip assays[23] and various mutant cell lines, we conducted a systematic analysis of TC-NER-dependent break induction following trabectedin treatment. In NER, damage is removed through a dual incision reaction, first by ERCC1-XPF on the 5' side to the lesion, followed by XPG on

**Fig. 1 | Trabectedin induces TC-NER-dependent DNA strand breaks in G1 cells.**
**a** Trabectedin and its DNA adduct structure (10.5452/ma-c4e6e) rendered using
PyMol. **b** HAP1 or U2OS WT, *XPC-*, *XPA-*, and *CSB*-KO cells were treated with tra-
bectedin or DMSO for 2 h, and colony counted after 8 days. Mean ± SEM of 3 and 2
biological replicates for HAP1 and U2OS, respecively (3 technical replicates per
experiment). *P*-values of ordinary two-way ANOVA with Dunnett's multiple com-
parisons test (between WT and mutants at each concentration) are provided.
**c** Scheme for assessing NER incision activity following DNA damage by alkaline
COMET chip assays. **d** HAP1 WT and **e** *XPA*-KO cells were arrested in G1 with pal-
bociclib and treated with trabectedin (50 nM, 2 h) and allowed to recover for up to
6 h with or without repair synthesis inhibitors (0.5 mM HU, 5 µM AraC). ssDNA
breaks were analyzed by alkaline COMET chip assays. **f** U2OS WT, **g** *XPC-*, **h** *CSB-* and
**i** *XPA*-KO cells were arrested in G1 with palbociclib (1 µM, 24 h) and treated with
trabectedin (50 nM, 2 h), and allowed to recover for up to 4 h with or without repair

synthesis inhibitors (1 mM HU, 10 µM AraC). ssDNA breaks were analyzed by alka-
line COMET chip assays. **d–i** Each dot represents DNA in tail (%) of a comet ana-
lyzed. Each box represents the mean value of DNA in tail (%) from all comets used in
all experiments. The number of comets used is provided above each box. An error
bar represents SD. **j** Summary and statistical analysis of COMET chip experiments in
HAP1 WT and *XPA*-KO cells. Mean ± SEM of 4 biological replicates. *P*-values of two-
tailed paired *t*-tests (between WT and XPA-KO at each recovery time) are provided.
Mean values of individual experiments (shown as boxes in panels **d**, **e**) served as the
input data. **k** Summary and statistical analysis of COMET chip experiments in U2OS
cells. Mean ± SEM of 4 (WT, *XPC*-KO), 3 (*CSB*-KO) biological replicates. No error bars
for *DDB2*-KO and *XPA*-KO (*n* = 1). *P*-values of ordinary two-way ANOVA with Dun-
nett's multiple comparisons test (between WT and *XPC*- or *CSB*-KO at each recovery
time) are provided. Mean values of individual experiments (shown as boxes in
panels **f–i**) served as the input data. Source data are provided as a Source Data file.

the 3' side to the lesion[24]. Our findings indicate that while XPF incision
occurs normally, the catalytic activity of XPG is inhibited by
trabectedin-DNA adducts. We leveraged this discovered mechanism of
trabectedin-induced SSB formation to map XPF-mediated incision
sites, revealing TC-NER activity on a genome-wide scale as well as
suggesting that XPF may cleave DNA in a sequence-specific way. Our
analysis showed that trabectedin induces SSBs predominantly on the
transcribed strand of active genes and to a lesser degree on the
opposite strand upstream of gene bodies due to divergent transcrip-
tion. Characterizing trabectedin-induced SSB landscapes across
diverse genotypes, we developed a robust approach – which we call
TRABI-seq – for probing TC-NER as well as transcription. The
mechanistic insight from our research could advance trabectedin's use
in precision oncology with trabectedin serving both as a drug and a
diagnostic for functional characterization.

## Results

### Trabectedin induces TC-NER-dependent DNA strand breaks in G1 cells

TC-NER deficiency renders cells resistant to trabectedin (Fig. 1a) but
sensitive to illudin S[5,6,19,21,22]. To confirm the reported cytotoxicity
profile, we treated TC-NER proficient or deficient HAP1, U2OS or XP
patient fibroblast cell lines with trabectedin. WT HAP1 and U2OS cells
as well as *XPA*-mutant patient cells (XP2OS; XP-A) complemented with
XPA-WT showed an IC50 in the range of 20–30 nM in clonogenic
survival assays (Fig. 1b, Supplementary Fig. 1a). GG-NER-deficient *XPC*-
knockout (-KO) HAP1 and U2OS cells were as sensitive as the WT cells,
while cells deficient in the TC-NER factor CSB were strikingly resistant
(Fig. 1b). Similarly, resistance was observed for *XPA*-KO HAP1, U2OS as
well as *XPA*-mutant XP2OS cells; these three cell lines are defective in
both NER pathways (Fig. 1b, Supplementary Fig. 1a). By contrast, and
consistent with previous reports[19,22], HAP1 cells deficient in CSB and
XPA displayed marked sensitivity to illudin S, a regular TC-NER sub-
strate, compared to WT and XPC-deficient cells (Supplemen-
tary Fig. 1b).

We hypothesized that the transcription-stalling trabectedin-DNA
adducts undergo an abortive TC-NER reaction that results in the for-
mation of persistent SSBs. To test this hypothesis, we set out to detect
SSBs using high throughput alkaline COMET chip assays[23]. In a stan-
dard NER reaction, for example following UV damage induction, SSBs
are transiently formed after incision of the damaged strand and before
completion of repair synthesis and ligation. Under normal conditions,
gaps formed by the removal of lesions are very short-lived and not
revealed by COMET assays (Fig. 1c, *scenario i*). By contrast, if NER
reactions are carried out in the presence of DNA repair synthesis
inhibitors AraC/HU, persistent SSBs are formed[25,26] (Fig. 1c, *scenario ii*).
We verified that these SSBs can be readily detected by COMET chip
assays in UV-exposed XP2OS cells and that comet tails were only
detected if cells expressed WT XPA and AraC/HU were added (Sup-
plementary Fig. 1c).

Having established a robust method for break detection, we
tested if trabectedin induced SSBs in a TC-NER dependent manner
(Fig. 1c, *scenario iii*). To reduce the background signal of breaks
formed during replication, we synchronized cells in G1 using the
CDK4/6 inhibitor palbociclib. We first treated HAP1 cells with tra-
bectedin, and measured time-dependent break formation by
COMET chip assays. Treatment with trabectedin alone resulted in
break formation about two-fold over background and the signal was
increased to about four-fold in the presence of Ara-C/HU (Fig. 1d).
No breaks were observed in XPA-deficient cells, showing that the
break formation was NER-dependent (Fig. 1e). In WT U2OS cells,
breaks were increased about four-fold 4 h after treatment with
trabectedin in the absence of DNA synthesis inhibitors (Fig. 1f).
Breaks were formed to a similar extent in XPC- and DDB2-deficient
cells, showing that these breaks are not caused by GG-NER (Fig. 1g,
Supplementary Fig. 1d). Breaks were absent in CSB- and XPA-
deficient cells, in line with TC-NER being responsible for break
formation (Fig. 1h, i). In TC-NER proficient U2OS cells, trabectedin-
induced breaks were again increased in the presence of AraC/HU
(from ~30% to ~50% DNA in the tail, albeit with a slightly higher
background signal in AraC/HU-treated cells) (Fig. 1f–i). We note that
the increase in SSBs following trabectedin treatment is higher in
U2OS compared to HAP1 cells (Fig. 1j, k). Although we do not know
the reason for this, it could be due to a difference in transcription
levels in the two cell lines. Since the increase of SSB formation is
statistically significant in both cell lines (Fig. 1j, k), we used both
interchangeably throughout the manuscript. Our results demon-
strate that trabectedin induces DNA breaks in a TC-NER-dependent
manner in G1 cells.

### XPF catalytic activity is necessary for trabectedin-induced DNA breaks

In NER, the incision 5' to the lesion by the XPF-ERCC1 endonuclease
precedes the incision 3' to the lesion by the XPG endonuclease[24].
Therefore, we hypothesized that the XPF-mediated incision, but not
the XPG-mediated incision, is required for trabectedin-induced
break formation. To test if the 5' incision is necessary for inducing
SSBs after trabectedin treatment, we used XPF-mutant-expressing
XP2YO patient fibroblasts either complemented with XPF-WT or
catalytically inactive XPF-D687A as well as HAP1 cells with an XPF-
D687A mutation at the endogenous locus (note that there was an
additional TCG(R) deletion at 701 in endogenous *XPF*, which might
have resulted in partial XPF protein degradation, Supplementary
Fig. 2a). We also used *ERCC1*-KO cells (Supplementary Fig. 2a), which
do not express functional XPF[27]. In the absence of XPF catalytic
activity, no breaks were observed in XP2YO cell lines expressing no
or mutated XPF when treated with UV and AraC/HU (Supplementary
Fig. 2b). The catalytic activity of XPF was necessary for the cyto-
toxicity of trabectedin as both *ERCC1*-KO and XPF-D687A cells were
resistant to trabectedin in HAP1 cells (Fig. 2a). Similarly, expressing

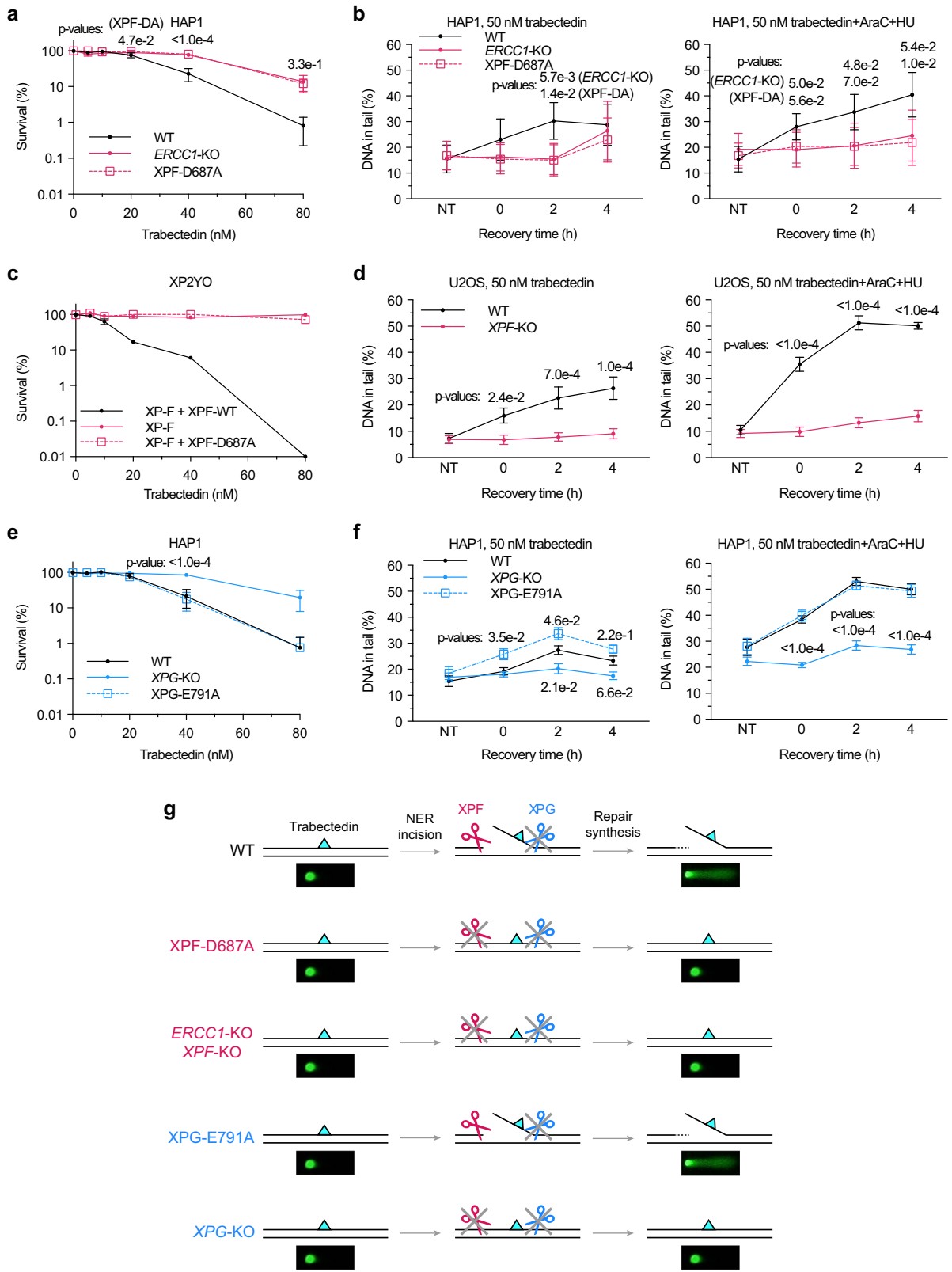

catalytically inactive XPF-D687A in XPF-mutant XP2YO cells did not restore sensitivity to trabectedin, while the expression of WT XPF did (Fig. 2c). Next, we used COMET chip assays to assess whether the observed survival pattern correlated with break formation. SSBs were observed in WT cells treated with trabectedin as expected, but no breaks were observed in *ERCC1*-KO or XPF-D687A cells at 2 h post trabectedin (Fig. 2b). With the caveat of the additional mutation in

the HAP1 XPF-D687A cells, our results indicate that the presence and nuclease activity of the ERCC1-XPF complex are needed for trabectedin-induced break formation (Fig. 2b and Supplementary Fig. 2c). We note that ERCC1 and XPF are critical for genomic integrity even in the absence of exogenous damage. This role of the proteins manifests itself more in ERCC1- and XPF-deficient HAP1 versus U2OS cells, considering that we observe residual sensitivity

**Fig. 2 | Trabectedin-induced DNA break formation and toxicity depend on the catalytic activity of XPF but not that of XPG. a** HAP1 WT, *ERCC1*-KO, and XPF-D687A cells were treated with trabectedin for 2 h and colonies counted after 7 days. Mean ± SEM of 5 (WT, XPF-D687A) and 3 (*ERCC1*-KO) biological replicates (3 technical replicates per experiment). Provided p-values are derived using ordinary two-way ANOVA with Dunnett's multiple comparisons test (between WT and *ERCC1*-KO or XPF-D687A at each concentration). **b** HAP1 WT, *ERCC1*-KO, and XPF-D687A cells were arrested in G1 with palbociclib and treated with trabectedin (50 nM, 2 h) and incubated for up to 4 h with or without repair synthesis inhibitors (0.5 mM HU, 5 μM AraC). ssDNA breaks were analyzed by alkaline COMET chip assays. Mean ± SEM of 3 biological replicates (Supplementary Fig. 2c). Provided *P*-values are derived using two-tailed paired t-test (between WT and *ERCC1*-KO or XPF-D687A at each recovery time). **c** XP2YO patient cells were treated with trabectedin for 2 h and colonies were counted after 8 days. Mean ± SEM of 2 biological replicates (3 technical replicates per experiment). **d** U2OS WT and *XPF*-KO cells were arrested in G1 with palbociclib and treated with trabectedin (50 nM, 2 h) and allowed to recover for up to 4 h with or without repair synthesis inhibitors (1 mM HU, 10 μM AraC). ssDNA breaks were analyzed by alkaline COMET chip assays.

Mean ± SEM of 4 biological replicates. *P* values were derived using ordinary two-way ANOVA with uncorrected Fisher's LSD (between WT and *XPF*-KO at each recovery time). **e** HAP1 WT, *XPG*-KO, and XPG-E791A cells were treated with trabectedin for 2 h and colonies counted after 7 days. Mean ± SEM of 4 biological replicates (3 technical replicates per experiment). Provided *P*-values are derived using ordinary two-way ANOVA with Dunnett's multiple comparisons test (between WT and *XPG*-KO or XPG-E791A at each concentration). **f** HAP1 WT, *XPG*-KO, and XPG-E791A cells were arrested in G1 with palbociclib (2 μM, 24 h) and treated with trabectedin (50 nM, 2 h). Cells were kept in G1 with or without repair synthesis inhibitors (0.5 mM HU, 5 μM AraC) and incubated for up to 4 h. ssDNA breaks were analyzed by alkaline COMET chip assays. Mean ± SEM of 5 biological replicates (Supplementary Fig. 3d–e). Provided *P*-values are derived using ordinary two-way ANOVA with Tukey's multiple comparisons test (between WT and *XPG*-KO or XPG-E791A at each recovery time). **g** A simplified schematic of assessing NER incision activity on trabectedin-induced DNA adducts in HAP1 WT, XPF-D687A, *ERCC1*-KO, XPG-E791A, and *XPG*-KO cells using alkaline COMET chip assays. Source data are provided as a Source Data file.

and breaks in HAP1 cells but not in U2OS cells lacking these proteins at later time points following trabectedin treatment (Fig. 2d).

### Evidence that trabectedin inhibits XPG catalytic activity

Since the second incision in NER is executed by XPG endonuclease, we tested whether the catalytic activity of XPG was needed for break formation by trabectedin. For this purpose, we generated HAP1 cells with catalytically inactive XPG-E791A (clones #22 and #45, Supplementary Fig. 3a), and confirmed that DNA breaks were induced following UV irradiation in the presence or absence of AraC/HU (Supplementary Fig. 3b), in line with reported effects of XPG with this mutation expressed in patient cell lines[23]. The addition of AraC/HU was not absolutely needed for the formation of SSBs following UV irradiation in cells expressing XPG-E791A, demonstrating that SSBs persist in the presence of catalytically inactive XPG (Supplementary Fig. 3b).

HAP1 WT or XP3BR (XP-G) patient cells complemented with XPG WT showed the expected level of sensitivity to trabectedin, while *XPG*-KO HAP1 or XP3BR (XP-G) patient cells were resistant (Fig. 2e, Supplementary Fig. 3c). Intriguingly, XPG-E791A expressing HAP1 (clone #22) or XP3BR cells were equally sensitive to trabectedin as WT cells, showing that the catalytic activity of XPG is not needed for trabectedin-induced toxicity. The difference in sensitivity of the *XPG*-KO and XPG-E791A cells can be explained with the observation that the presence, but not catalytic activity of XPG, is needed for the XPF incision to occur[24,28].

To determine if DNA breaks are responsible for toxicity observed in XPG-E791A cells, we conducted alkaline COMET chip assays with XPG WT, *XPG*-KO, and XPG-E791A cells. There were no SSBs in *XPG*-KO cells at 2 h post trabectedin, whereas breaks were induced in XPG-E791A cells, seemingly at even higher levels than in WT cells in the absence of DNA synthesis inhibitors (Fig. 2f, left; Supplementary Fig. 3d–e). In the presence of DNA synthesis inhibitors, break formation was increased, and found to be at the same levels in the XPG WT and XPG-E791A cells (Fig. 2f, right). The fact that trabectedin-induced break formation in WT and E791A cells is very similar supports the model that trabectedin-DNA adducts block XPG incision.

Our studies suggest that trabectedin-induced break formation and toxicity depend on the catalytic activity of XPF, but not that of XPG (Fig. 2g). In this way, only one NER incision is made, leading to a persistent SSBs with a free 3'-OH upstream of the trabectedin-DNA adduct.

### Trabectedin-induced DNA breaks can be mapped genome-wide

The COMET chip experiments have revealed a mechanism responsible for TC-NER- and trabectedin-induced SSB formation but provided no information on where in the genome these breaks occur. Are trabectedin-induced SSBs evenly distributed throughout the genome,

do they occur in specific genomic locations, or do they perhaps target oncogenes upregulated in certain tumors?

We hypothesized that our mechanistic insight would provide a basis for an approach to reveal trabectedin-induced SSBs with single-nucleotide resolution and in a genome-wide fashion. Specifically, we aimed to map the persistent ERCC1-XPF-dependent SSBs upstream of the drug-DNA adduct, employing the recently developed method of genome-wide ligation of severed 3'-OH ends followed by sequencing, GLOE-Seq[29]. We introduced an upgrade to the method that provides a more balanced representation of SSB signals and enables filtering out PCR amplification duplicates of sequencing reads. We labeled DNA fragments originating from DNA breaks first with a biotinylated proximal adapter for enrichment and subsequently with a distal adapter containing a unique molecular identifier (UMI) of randomized nucleotides to reflect the abundance of a given DNA break at a certain genomic position across the population of cells (Fig. 3a). As a positive control for the upgraded method, we treated U2OS WT DNA with the Nb.BsrDI nicking endonuclease that introduces SSBs before CATTGC sequences and confirmed that GLOE-Seq maps breaks at this sequence with at least 82–84% precision (true positives vs true plus false positives), i.e., the fraction of reads revealing the CATTGC pattern out of all reads in a sample, (Supplementary Fig. 4a) and 87–88% sensitivity (true positives vs true positives plus false negatives), i.e., the fraction of identified Nb.BsrDI sites in the human reference genome (Supplementary Fig. 4b). This analysis showed that the upgraded method performed on the human genome at least as well as the original GLOE-Seq protocol, which was validated in the same way in the yeast genome[29].

Using the upgraded GLOE-Seq method, we set out to reveal the genomic positions of the ERCC1-XPF-mediated trabectedin-induced breaks. We determined genome-wide profiles of DNA breaks in the TC-NER-proficient WT U2OS cell line, the corresponding TC-NER-deficient *CSB*-KO, global-genomic-NER-deficient *XPC*-KO and fully NER-deficient *XPA*-KO cells. Cells were treated with trabectedin (50 nM) or vehicle control using the same conditions as in the COMET chip assays (2h treatment and 2h recovery). We found abundant trabectedin-induced DNA breaks in the TC-NER-proficient cells (U2OS WT and *XPC*-KO), while we did not obtain strong evidence for the genome-scale elevation of trabectedin-induced DNA breaks in the TC-NER-deficient cells (*CSB*-KO and *XPA*-KO), as shown in detail for Chromosome 19 (Fig. 3b), in an overview for the whole genome (Supplementary Fig. 4c) and related quantification (Supplementary Fig. 4d–e). These GLOE-Seq maps were consistent across biological replicates (Spearman correlation coefficient mostly around 0.9) (Supplementary Fig. 4f). We observed that the genome-wide profile of DNA breaks in the drug-exposed U2OS WT

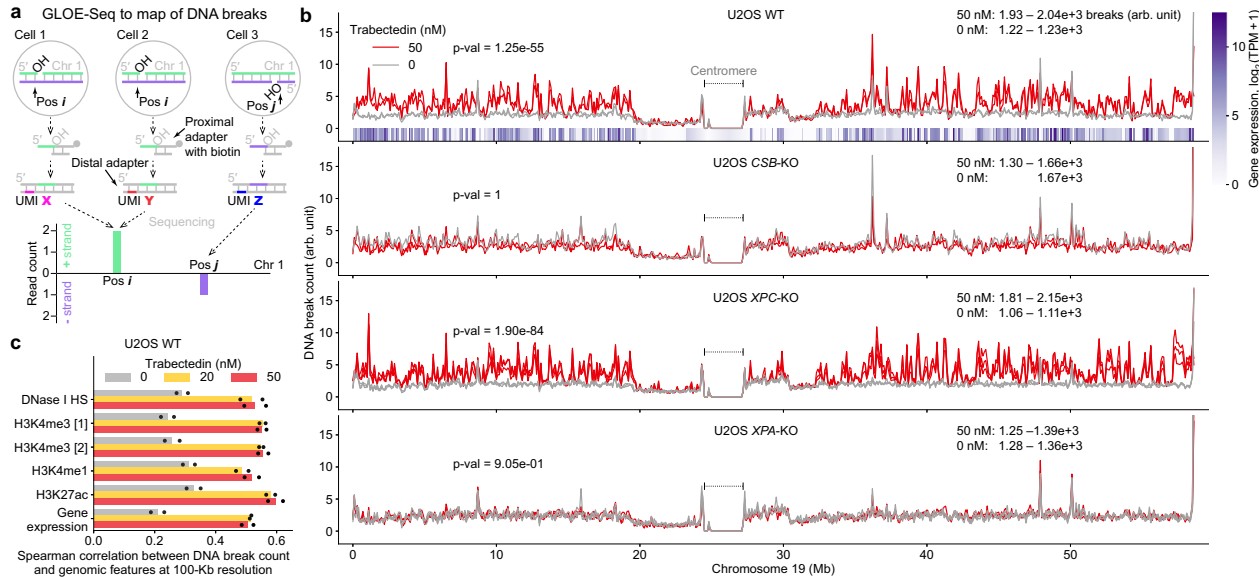

**Fig. 3 | Trabectedin-induced DNA breaks are mapped genome-wide with upgraded GLOE-Seq. a** Principle of GLOE-Seq that maps DNA breaks in a genome-wide and strand-specific fashion and provides an estimate of the frequency of individual breaks in a population of cells. The distal adapter with a unique molecular identifier (UMI) is an upgrade from the original GLOE-Seq protocol[29]. Pos: position, chr: chromosome, OH: free 3' hydroxyl. **b** DNA break count along chromosome 19 in 4 cell lines after 2 h exposure to trabectedin or vehicle (DMSO) and subsequent 2 h recovery. Solid lines: individual biological replicates, 2 for 50 and 0 nM drug in WT, 50 nM in *CSB*-KO; 1 for 0 nM in *CSB*-KO; 3 for 50 and 0 nM in *XPC*-KO and *XPA*-KO. Vertical bar heatmap: gene expression level in unexposed U2OS WT. All data are shown per 100-Kb bin. We summed DNA-break counts within this chromosome and provided the min-max range of this value across replicates for

either treatment condition (shown also in Supplementary Fig. 4d). *P*-value: Mann-Whitney U test with the one-sided alternative hypothesis that the trabectedin-treatment-related distribution is stochastically greater than the control distribution (see more details in Supplementary Fig. 4e). TPM, transcripts per million transcripts. Arb. unit, arbitrary unit: Methods describe DNA break count normalization. **c** Genome-wide correlation of DNA break count with the abundance of DNase I hypersensitivity (HS) sites (transcriptional activity), H3K4me3 (active gene promoters), H3K4me1 (active enhancers) and H3K27ac (active promoters and enhancer) as well as gene expression. Bars: mean, markers: biological replicates (n = 2) of break mapping. *N* = 28,513 genomic bins to compute the correlation. Source data are provided as a Source Data file.

---

had a marked correlation with transcriptionally active regions of chromatin (DNase I hypersensitivity sites), epigenetic marks of active promoters and enhancers (H3K4me3, H3K4me1, H3K27ac) as well as gene expression levels (Fig. 3c).

## Trabectedin-induced DNA break counts correlate with gene expression levels

Therefore, we next focused our analysis on genes, and plotted the DNA break counts (Y axis) for transcribed (green) and non-transcribed strands (purple) versus gene expression levels (X axis) (Fig. 4a–d). We normalized the DNA break count data in each experiment by fixing the average DNA break count for transcribed (antisense) strand of unexpressed genes to 1. Gene DNA counts were normalized by gene length as longer genes naturally have more breaks (see normalization formulas in Methods). We discovered that in U2OS WT cells, for the vast majority of unexpressed genes and genes up to the 50th percentile of gene expression, the DNA break count in the transcribed strand was below 4. By contrast, a more pronounced increase of DNA break count was observed at higher gene expression levels (Fig. 4a, heatmap). When we stratified the genes based on expression levels, we found that the average break count in the transcribed strands of the top 5% expressed genes was around 9-fold higher than in unexpressed genes, whereas DNA break count in the non-transcribed (sense) strand did not similarly depend on gene expression (Fig. 4a, boxplot). These data show that trabectedin induces DNA breaks preferentially in transcribed strands of highly expressed genes. We observed a similar correlation in TC-NER-proficient *XPC*-KO cells (Fig. 4c) but not in TC-NER-deficient *CSB*-KO (Fig. 4b) or *XPA*-KO cells (Fig. 4d), in agreement with COMET chip assays (Fig. 1k). In DMSO-treated control, the break counts were the same throughout gene

expression levels for all 4 U2OS cell lines used (Supplementary Fig. 5a–d). Similar patterns were observed in HAP1 WT cells (Supplementary Fig. 5e–f), although the fold change in break formation throughout the gene expression tiers was lower than in U2OS cells, which is consistent with the lower fold increase in the DNA break formation in HAP1 versus U2OS cells determined in the COMET chip assay (Fig. 1j–k).

We found our assay to be robust, detecting a near-identical break count pattern over biological replicates (Fig. 4e–h, Supplementary Fig. 5g–h) and a lower break count at a lower trabectedin concentration of 20 nM versus 50 nM (Fig. 4e). Interestingly, we identified a very minor, but reproducible expression-dependent trabectedin-induced break formation on the non-transcribed strand only in TC-NER-proficient U2OS WT and *XPC*-KO cells (Fig. 4e–h, inner plots) as well as HAP1 WT (Supplementary Fig. 5g, inner plot). We could also detect that in the fully NER-deficient *XPA*-KO cells, there was still a low level (up to 2-fold) of gene expression-dependent trabectedin-induced break formation (Fig. 4h, inner plots), while in the TC-NER-deficient *CSB*-KO, no trabectedin-induced breaks were formed (Fig. 4f). We currently do not understand how these rare XPA-independent breaks are formed but favor the hypothesis that they do not represent NER in the absence of XPA. Besides, we observed a modest correlation between break counts and gene expression without trabectedin exposure, with this correlation for the transcribed strand being consistently higher than for non-transcribed strand (Supplementary Fig. 5h, gray bars), which may indicate endogenous break formation induced by transcription. While we do not currently have a biological explanation for trabectedin-induced break formation in the non-transcribed strands and in the *XPA*-KO cells, these observations demonstrate the quantitative nature of our assay.

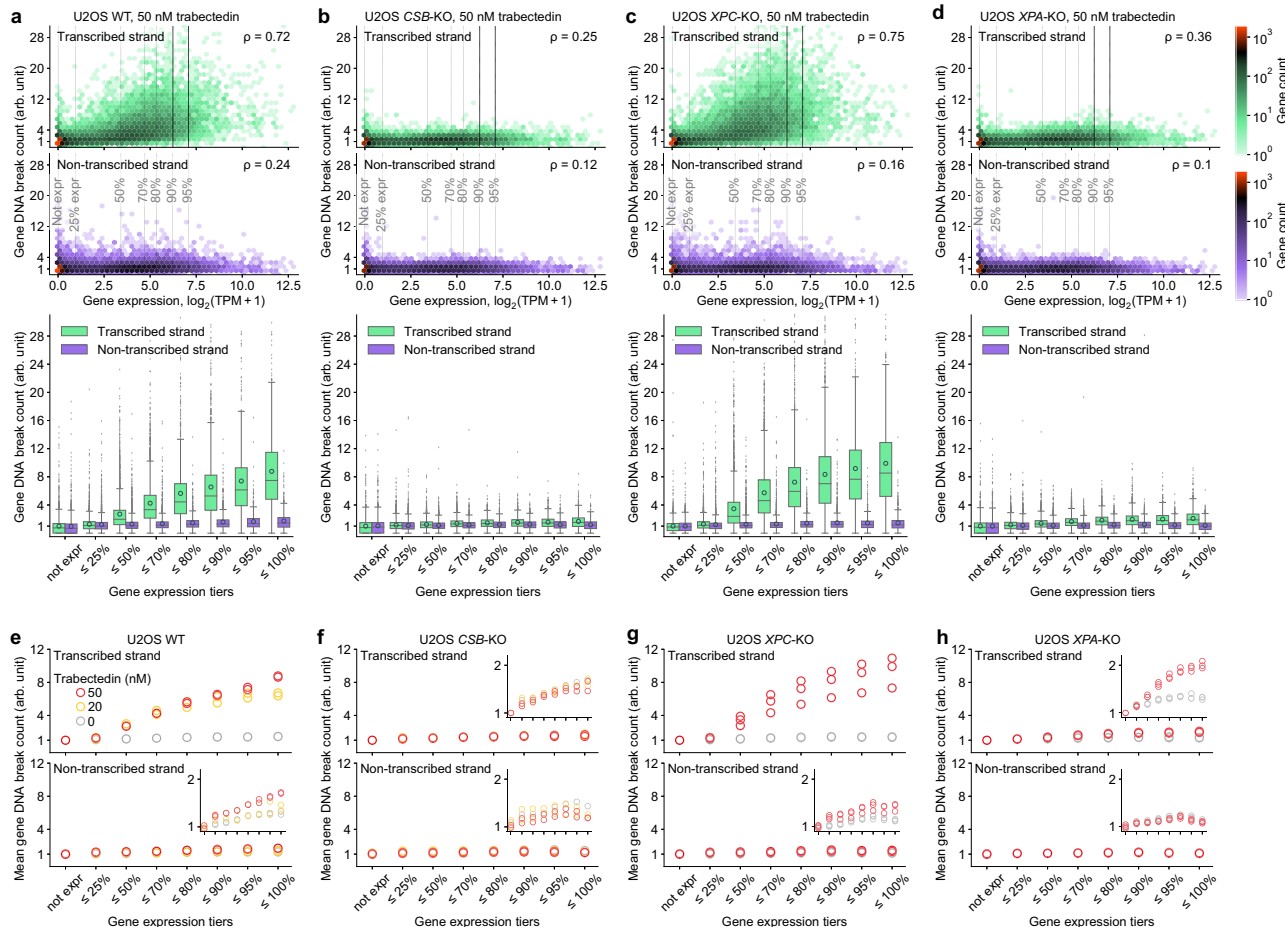

**Fig. 4 | Trabectedin-induced DNA break counts correlate with gene expression levels.** DNA break count on each strand of protein-coding genes in U2OS WT (**a**), *CSB*-KO (**b**), *XPC*-KO (**c**) and *XPA*-KO (**d**) after 2 h exposure to trabectedin and subsequent 2 h recovery versus gene expression level in unexposed U2OS WT. $\rho$: Spearman correlation coefficient calculated for continuous (not tiered) data of gene expression and DNA break count (correlation analysis for all replicates in Supplementary Fig. 5h), $n = 16,740$. Heatmaps: the color of a hexagonal bin reflects the gene count. TPM, transcripts per million transcripts. Box plots: gene expression was categorized in tiers via percentiles shown by vertical lines in the heatmaps; boxes are interquartile ranges, internal horizontal lines are medians, circular markers are means, whiskers extend to the furthest datapoint within 1.5x interquartile

range, datapoints beyond are shown as small markers. Number of genes per tier and number of genes beyond the maximal y-axis value are provided in **Methods**. Presented data: one biological replicate of drug exposure per cell line. Mean DNA break count of protein-coding genes in U2OS WT (**e**), *CSB*-KO (**f**), *XPC*-KO (**g**) and *XPA*-KO (**h**) after 2h exposure to trabectedin or vehicle and subsequent 2h recovery versus gene expression level in unexposed U2OS WT. Presented data: multiple biological replicates. Gene expression tiers are described in **a**–**d**. Number of genes per tier are provided in Methods. The internal plots zoom in on the y-axis. a-h: arb. unit, arbitrary unit: Methods describe DNA break count normalization. Source data are provided as a Source Data file.

## Divergent transcription is detected by trabectedin-induced DNA breaks

Having observed that trabectedin induces damage preferentially in transcriptionally active genes, we further zoomed in on the DNA break distribution throughout gene bodies and adjacent regions in these genes. In U2OS WT cells treated with trabectedin (50 nM), the profiles of individual genes with the highest damage showed that DNA breaks were most abundant in the transcribed strand right downstream of the transcription start site (TSS) (Fig. 5). Evaluating the damage in all highly expressed genes together, we found that the mean DNA-break count also peaked right downstream of TSS in the transcribed strand and gradually decreased throughout the gene body, being 3 to 4-fold lower at the transcription end site (TES) (Fig. 6a, green traces). In line with the data of Fig. 4a–d, this DNA break pattern was absent in TC-NER-deficient U2OS *CSB*-KO (Fig. 6b) and *XPA*-KO cells (Fig. 6d) treated with trabectedin as well as in all untreated cells (Supplementary Fig. 6a–d,f). It was however very similar in trabectedin-treated TC-NER-proficient U2OS *XPC*-KO (Fig. 6c) and HAP1 WT cells (Supplementary Fig. 6e). The peak values of DNA breaks were found around 1–2 kb after the TSS

independently of the gene length (Fig. 6e,f, left panels). We surmise that the higher break formation closely downstream of the TSS is due to the stalling of RNA polymerase at the first trabectedin-DNA adducts early in the gene and the consecutive inhibition of the transcription and TC-NER activity. Elevated break levels were also detected after TES, potentially reflecting alternative longer transcripts (Fig. 6a,c, Supplementary Fig. 6e).

Interestingly, we found another peak of DNA break counts a few hundred nucleotides upstream of the TSS on the non-transcribed strand, with the break count gradually decreasing yet not reaching background levels even 5 Kb upstream of the TSS (Fig. 6a, c, e, f and Supplementary Fig. 6e). The frequency of the DNA breaks in the non-transcribed strand increased only further than 100 nucleotides upstream of the TSS, making this 100-nucleotide region of the promoter devoid of trabectedin-induced breaks on both DNA strands (Fig. 6e, f, right panels). These break patterns appear to be in line with the phenomenon of divergent transcription, i.e., transcription on both DNA strands occurring in opposite directions, which is observed in most mammalian

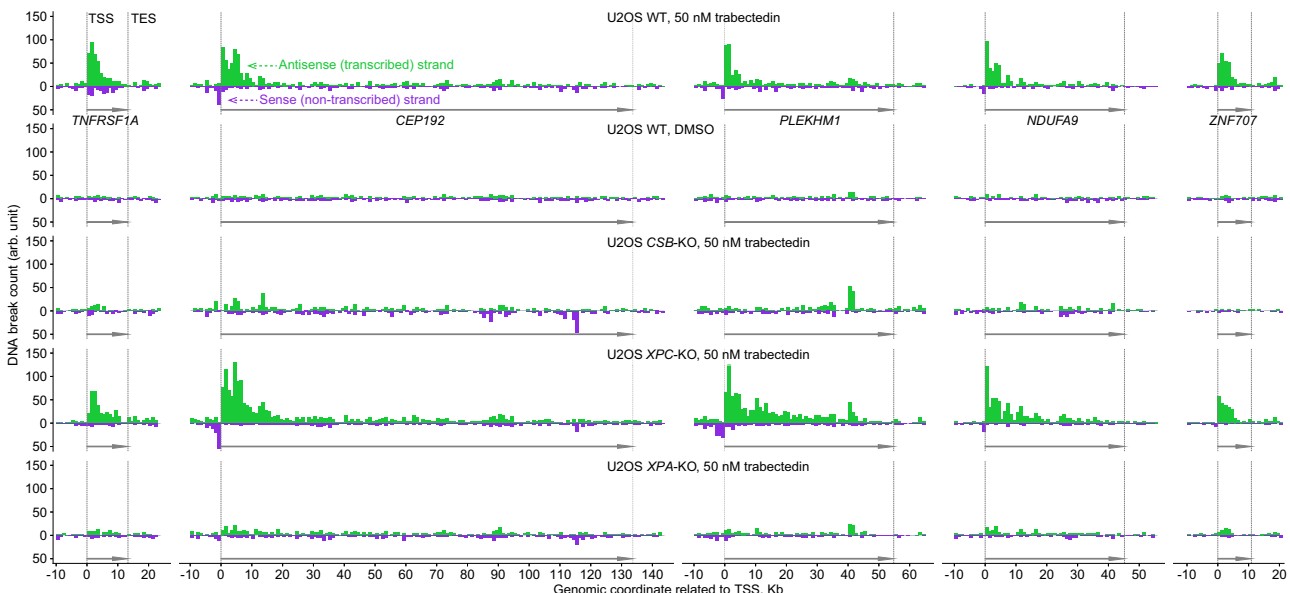

**Fig. 5 | DNA-break profiles of genes with highest trabectedin-induced damage.** Strand-specific profiles of individual genes in the indicated cell lines after 2h exposure to trabectedin or DMSO and subsequent 2h recovery. Genomic coordinates were adjusted such that TSS is zero and gene directionality is rightward. Bin length: 1 kilobase. Bar: mean across biological replicates, $n = 2$ for WT 50 nM drug, WT DMSO, *CSB*-KO 50 nM, $n = 3$ for *XPC*-KO 50 nM and *XPA*-KO 50 nM. Kb: kilobase. We present protein-coding genes with highest trabectedin-induced damage within 5 Kb downstream of TSS in U2OS WT. *PLEKHM1* is in top 50% expressed genes of U2OS WT, the other genes are in top 30%. Source data are provided as a Source Data file.

promoters[30–33], where the absence of strong TATA elements may lead to transcription bidirectionality[34]. We further found that the trabectedin-induced break count upstream of the TSS in the non-transcribed strand is increasing with the gene expression level (Fig. 6g, h, purple; Supplementary Fig. 6g–h), similarly to the break count downstream of the TSS in the transcribed strand (Fig. 6g, h, green). However, the TTS-upstream breaks are on average 2-fold less abundant than the TSS-downstream breaks in genes after the 25th percentile of gene expression (Fig. 6g–h), which indicates that divergent transcription activity is about a half of that in the gene body averaged over the entire genome. Overall, trabectedin-induced break sequencing – a method we call TRABI-seq – reports on various types of transcription, including divergent transcription, and demonstrates that TC-NER can also occur in the intergenic space.

### TRABI-seq may reveal a sequence preference for XPF incision

As the COMET assay experiments showed that trabectedin-induced DNA breaks are caused by ERCC1-XPF activity (Fig. 2b), locating these breaks in TRABI-seq provides an opportunity to explore potential sequence preferences of the incision activity of this endonuclease. To analyze whether the ERCC1-XPF has a sequence preference, we focused on the top 30% expressed genes and examined DNA breaks in the regions most affected by the drug, namely, the transcribed strands of the genes and 5 Kb of the non-transcribed strands upstream of the TSS, where any sequence preference would not be obscured by background signals (Fig. 6a, c). We aligned the sequence contexts of these breaks and discovered that there is an enrichment of guanine at the second position downstream of DNA breaks (Fig. 6i, j). According to these data, ERCC1-XPF may prefer a guanine (or complementary cytosine) one base downstream of the incision site. No overrepresented nucleotides were found further downstream of the break location (Fig. 6i, j), or in TC-NER deficient *CSB*-KO or *XPA*-KO cells (Supplementary Fig. 6i, j). Thus, by providing evidence for ERCC1-XPF DNA sequence specificity, TRABI-seq serves as a tool to study the fundamental properties of NER.

## Discussion

DNA-damaging agents are crucial therapeutics used in the treatment of most cancers, however, innate or acquired resistance of tumors to these agents is a major limitation. New developments to increase the efficacy of anticancer therapy rely on precision oncology where specific drugs are matched with the genetic profile of tumors. As successful clinical examples, cisplatin and poly (ADP-ribose) polymerase inhibitors (PARPi) target tumors with defects in *BRCA1/2* and other genes involved in mediating homologous recombination and counteracting replication stress. Similarly, tumors with defects in NER are also hypersensitive to cisplatin, while high NER activity is associated with resistance. Recent studies have additionally put forward irofulven, a derivative of illudin S, as an agent to selectively target tumors with deficiencies in TC-NER[35,36]. The therapeutic action of PARP inhibitors, cisplatin and irofulven thus relies on synthetic lethality, i.e., tumor vulnerability due to a defect in the repair pathway counteracting the toxic effects of the drug.

Here we investigated trabectedin, a drug that shows an opposite mode of action, namely being more toxic to cells with high repair capacity. Trabectedin is a promising drug to treat tumors with high TC-NER activity or, generally, with intact DNA repair machinery. Additionally, trabectedin may be valuable in overcoming the NER-mediated resistance of tumors to cisplatin. To further its use in precision medicine, we set out to study its mechanism of NER-induced toxicity. Using highly sensitive COMET chip assays, we show that trabectedin induces persistent DNA breaks in a TC-NER-dependent manner in cells that are hypersensitive to the drug. Our data suggests a model that the trabectedin-DNA adducts block the incision of the XPG endonuclease, which results in persistent XPF-mediated breaks (Fig. 7). We mapped those breaks genome-wide and showed that they form predominantly in highly transcribed genes and their upstream regions in association with divergent transcription. This approach of break sequencing, which we call TRABI-Seq, is informed by the uncovered mechanism of trabectedin and provides opportunities to study TC-NER and profile tumor vulnerability by mapping TC-NER activity.

The discovered mechanism of trabectedin action is in line with the prevailing model for NER dual incision where XPF cuts before XPG[24,37].

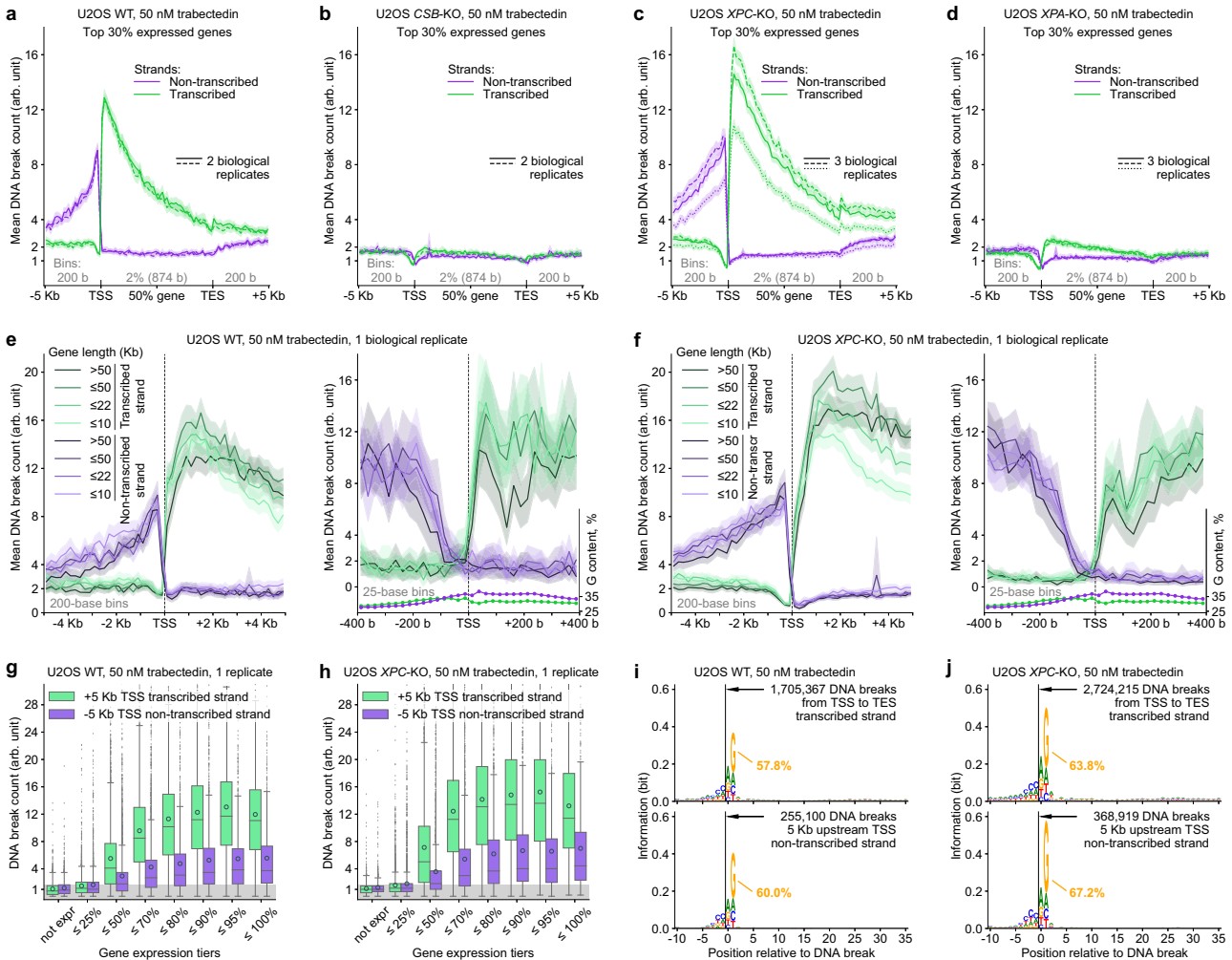

**Fig. 6 | TRABI-seq detects divergent transcription and provides evidence for *XPF* sequence preference.** Strand-specific profile of the mean DNA break count and its 95% c.i. (shade) throughout the gene body and adjacent regions in U2OS WT (**a**), *CSB*-KO (**b**), *XPC*-KO (**c**) and *XPA*-KO (**d**) after 2h exposure to trabectedin and subsequent 2h recovery. *n* = 4425 protein-coding genes (top 30% expressed in unexposed U2OS WT) are considered to compute the means and c.i. Solid, dashed, and dotted curves: means of different biological replicates. Strand- and gene-length-specific profile of the mean DNA break count and its 95% c.i. (shade) in the ±5 kilobase (Kb) proximity of TSS in TC-NER proficient cell lines U2OS WT (**e**) and *XPC*-KO (**f**), zooming out (left panel) and in (right). The same gene set as in **a**–**d**. Methods provide gene numbers per gene-length group. DNA break count in two branches of divergent transcription in U2OS WT (**g**) and *XPC*-KO (**h**) versus gene expression. +5 Kb: within 5 Kb downstream of; −5 Kb: within 5 Kb upstream of. The plots are built analogously to Fig. 4a–d (lower panels). Supplementary Fig. 6g–h presents respective correlation analysis for all replicates. Gray band: endogenous DNA breaks not caused by trabectedin treatment (upper quartile of DNA break count in unexpressed genes); this threshold shows that around 25% (lower boundary of boxes) of highly expressed genes may not have trabectedin-induced breaks upstream of the TSS. Sequence logos around DNA breaks in U2OS WT (**i**) and *XPC*-KO (**j**). We considered DNA breaks located in the indicated regions of the gene set used in **a**–**f**. The percentage of G at position 1 (+2 relative to the break) is shown. Data: all biological replicates united per cell line. Supplementary Fig. 6i–j: analogous analysis for TC-NER-deficient cell lines. **a**–**f**: bin sizes are absolute (a base number) or relative (a percentage of gene length; the corresponding average base number indicated in parentheses). **a**–**h**: arb. unit: Methods describe DNA break count normalization. **a**–**j** TSS and TES: transcription start and end sites; Kb: kilobase; b: base. Source data are provided as a Source Data file.

Our data shows that during TC-NER of trabectedin, 5′ incision by XPF occurs normally, while 3′ incision by XPG is blocked. This observation is consistent with the fact that the incisions in NER are asymmetric with respect to the position of lesions being located much closer to the 3′ incision site by XPG (2–8 bases) than to the 5′ incision site by ERCC1-XPF (15–24 bases)[38–40]. Furthermore, structural models indicate that the bulky trabectedin DNA adduct is oriented toward the 3′ incision site, where it may interfere with the catalytic activity of XPG[9].

A potential application of trabectedin in precision medicine may involve using TRABI-Seq to profile TC-NER activity in tumors. This assay, validated across a range of knockout cell lines (Fig. 4e–h, Supplementary Fig. 5g–h), offers a robust and sensitive method for functionally identifying TC-NER deficiencies in tumors. Such deficiencies could, for instance, indicate a candidate tumor for the drug irofulven,

which is synthetic lethal in combination with inactive TC-NER[35,36]. TRABI-seq-based reporting of high TC-NER activity in tumors would, on the contrary, suggest trabectedin as a therapeutic. Along with profiling TC-NER activity, TRABI-seq could also characterize tumors with respect to their oncogene expression patterns (Supplementary Fig. 7) as gene break counts strongly correlate with gene expression (Fig. 4a, e). Interestingly, a recent report used a biotinylated derivative of lurbinectedin, a close relative of trabectedin, to map lurbinectedin-DNA adducts in a Chip-seq approach[41]. While this method can locate the adducts to the promotors of tumor-driving genes, TRABI-seq reports also on the location of cytotoxic breaks, TC-NER and gene expression.

The persistent DNA breaks resulting from trabectedin adduct-induced recruitment and consecutive abortion of TC-NER machinery

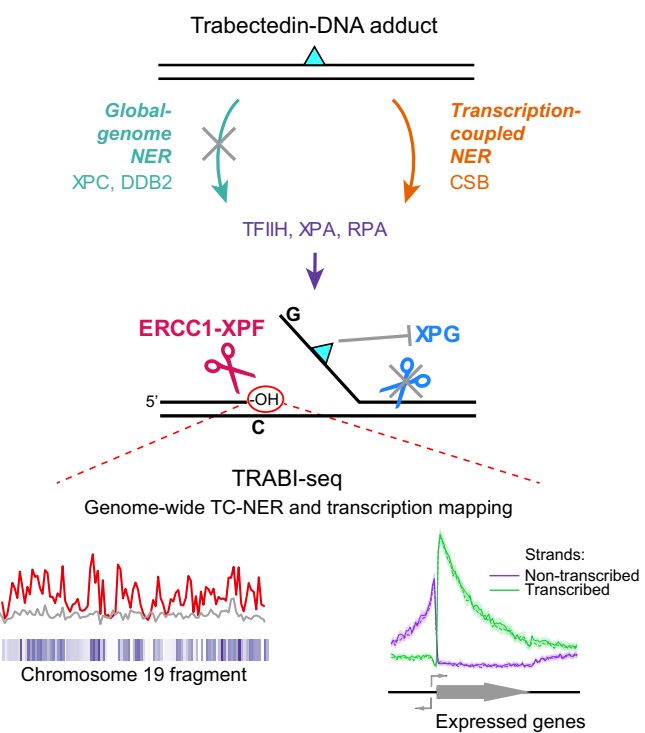

Trabectedin-DNA adduct

**Fig. 7 | Summary of the mechanism of trabectedin-induced TC-NER-mediated SSB formation and toxicity, and the development of TRABI-Seq.** Trabectedin-DNA adducts are exclusively recognized by TC-NER, and not by GG-NER. The adducts then block the incision of the XPG endonuclease, causing persistent XPF-mediated breaks. Mapping those breaks in a genome-wide fashion displays where TC-NER is active: predominantly in highly transcribed genes and their upstream regions due to a link to divergent transcription.

offer a unique opportunity to study this pathway of DNA repair. We note that we have not yet fully explored the nuances of how various TC-NER genes affect trabectedin-induced toxicity and lesion processing. For example, it is known that CSB is involved in RNA polymerase II degradation and repair, while other TC-NER specific factors, such as UV-SSA, specifically contribute to repair. It will be intriguing to explore in more detail how these properties influence cellular responses to trabectedin[13,14]. By using TRABI-Seq, we surprisingly discovered that TC-NER is active beyond genes, likely due to divergent transcription branch oriented in the direction upstream of promoters and using the non-transcribed (from the gene perspective) DNA strand as a template (Fig. 6a, c, Supplementary Fig. 6e)[30–33]. Another intriguing aspect of TRABI-Seq data is evidence for DNA sequence guiding XPF activity (Fig. 6i-j), which requires further investigation but is in line with recent findings regarding another DNA-repair endonuclease, MRE11-RAD50-XRS2, which was also found to cleave with a sequence preference[42].

In conclusion, we uncovered that trabectedin blocks one of two incision reactions in TC-NER and used this finding to map TC-NER and gene expression activity on genome-wide scale. This insight and developed techniques will advance investigations into the mechanism of trabectedin toxicity and inform its use in precision oncology.

## Methods
### Trabectedin, illudin S and antibodies
Trabectedin used in survival and COMET experiments was from Tecoland and in GLOE-seq experiments from Lucerna-Chem AG. Illudin S was from MGI Pharma. β-actin (catalog no. MA5-15739) was from Invitrogen. XPF antibodies were from Abcam (Catalog no. ab76948) and Santa Cruz (Catalog no. sc-136153). XPG was from Bethyl (catalog no. A301-484A). ERCC1 (D-10) was from Santa Cruz Biotechnology (Catalog no. sc-17809). Goat anti-rabbit IgG (Catalog no. ADI-SAB-300-

J) and goat anti-mouse IgG F(ab')2 (Catalog no. ADI-SAB-100-J) were from Enzo Life Sciences.

### Cell lines
CML derived human HAP1 wild-type, *XPC*-, *XPA*-, *CSB*- and *XPG*-KO cells were from Horizon Discovery. HAP1 XPG-E791A, XPF-D687A, and *ERCC1*-KO cells were generated with CRISPR-Cas9 (see below). SV40-transformed human XP-F fibroblasts XP2YO (XPF-deficient, GM08437), XP2YO complemented with wild-type XPF or mutant XPF-D687A and SV40-transformed human XP-A fibroblasts XP2OS and XP2OS complemented with wild-type XPA were previously reported[24,43]. SV40-transformed human XP-G fibroblasts XP3BR (XPG-deficient) were from Kaoru Sugasawa (Kobe University) and were complemented with wild-type XPG or XPG-E791A by lentiviral transduction as previously decribed[24]. Lentivirus was prepared by transfecting 293 T cells with 0.75 μg of pWPXL vector (XPG wild type or E791A cDNA), 2.25 μg of pMD2.G envelope plasmid, and 2.25 μg of psPAX2 packaging plasmid, using Lipofectamine 3000 (catalog no. L3000001, Thermo Fisher Scientific). The virus was harvested after 24 h. XP3BR cells, seeded at 50% confluency, were infected with the virus at an MOI of 2 for 24 h. Cells were subsequently cultured as described below. U2OS WT, *XPC*-KO, *CSB*-KO, and *XPA*-KO cells were from Martijn S. Luijsterburg (Leiden University Medical Center)[44], and U2OS *DDB2*-KO and *XPF*-KO cells were from Hannes Lans, Jurgen A. Marteijn and Wim Vermeulen (Erasmus University Medical Center, Rotterdam)[45,46].

### Cell culture
HAP1 cells were maintained in IMDM containing 4.5 g/L glucose with 10% fetal bovine serum, 1% penicillin-streptomycin, and GlutaMAX™ at 37 °C with 5% $CO_2$ (Gibco). XP2YO, XP2OS, and XP3BR cells were maintained in DMEM containing 4.5 g/L glucose and 2mM l-glutamine (Cytiva) with 10% fetal bovine serum (FBS, Millipore) and 1% penicillin-streptomycin (P/S, Gibco) at 37 °C with 5% $CO_2$. For survival and COMET chip experiments, U2OS cells were maintained in DMEM containing 4.5 g/L glucose and 2mM l-glutamine (Cytiva) with 10% fetal bovine serum (FBS, Millipore) and 1% penicillin-streptomycin (P/S, Gibco) at 37 °C with 5% $CO_2$. For TRABI-seq experiments with U2OS WT, *XPC*-KO, *CSB*-KO and *XPA*-KO cells were maintained in McCoy's 5a modified cell culture medium (Gibco) at 37 °C with 5% $CO_2$.

### Generation of mutant HAP1 cell lines with CRISPR-Cas9
For XPG-E791A and XPF-D687A HAP1 cell lines: sgRNAs, single-stranded oligodeoxynucleotides (ssODNs), purified S.p.Cas9 (catalog no. 1081058), and Alt-R HDR Enhancer (catalog no. 1081072) were purchased from IDT. For XPG-E791A knock-in, the following sequences were used; sgRNA (sequence: CACTGCGCCTCTGCTTCCAT): mC*mA*mC*rUrGrCrGrCrCrUrCrUrGrCrUrUrCrCrArUrGrUrUrUrUrArGrArGrCrUrArGrArArArUrArGrCrArArGrUrUrArArArArUrArArGrGrCrUrArGrUrCrCrGrUrUrArUrCrArArCrUrUrGrArArArArArGrUrGrGrCrArCrCrGrArGrUrCrGrGrUrGrCmU*mU*mU*rU, ssODN:/AlT-R HDR1/C*C*TGCGCCTGTTCGGCATTCCCTACATCCAGGCTCCCATGGAAGCGGCCGCGCAGTGCGCCATCCTGGACCTGACTGATCAGACTTC*C*G/AlT-R-HDR2/. For XPF-D687A knock-in, the following sequences were used; sgRNA (sequence: CAATGTCAATGCCCCGACGA): mC*mA*mA*rUrGrUrCrArArUrGrCrCrCrCrGrArCrGrArGrUrUrUrUrArGrArGrCrUrArGrArArArUrArGrCrArArGrUrUrArArArArUrArArGrGrCrUrArGrUrCrCrGrUrUrArUrCrArArCrUrUrGrArArArArArGrUrGrGrCrArCrCrGrArGrUrCrGrGrUrGrCmU*mU*mU*rU, ssODN: /AlT-R-HDR1/T*G*GCCAGGAACAGAATGGTACACAGCAAAGCATAGTTGTGGCAATGCGTGAATTTCGAAGTGAGCTTCCATCTCTGATCCATCGTCGGGGCATTGACATTGAACCCGTGACTTTAGAGGTTG*G*A/AlT-R-HDR2/. CRISPR ribonucleoprotein (RNP) complexes were transfected into HAP1 cells using CRISPRMAX™ (catalog no. CMAX0001, Thermo Fisher Scientific). 0.5 μL of sgRNA (3 μM), S.p.Cas9 (3 μM), and Cas9 Plus Reagent (0.5 μL, a part of CRISPRMAX™) were mixed with 10.5 μL Opti-MEM™ (catalog no.

31985062, Thermo Fisher Scientific) and incubated at room temperature (RT) for 10 min to make ribonucleoprotein complex (RNP). 120 nM of RNP (12.5 μL) was mixed with 0.3 μL CRISPRMAX™ Reagent (a part of CRISPRMAX™) and 5 μL Opti-MEM™ and incubated at RT for 10 min. The RNP/liposome complex was added into a well of 96-well plate with 0.4 X $10^4$ cells in total IMDM growth medium (125 μL), followed by addition of 1.25 μL 3 mM Alt-R HDR Enhancer. Cells with RNP complex were incubated at 37 °C with 5% $CO_2$ for 72 h. The mutant genome loci were amplified with PCR primers to validate the mutation (XPG: Forward-CCCTGGGGGAATGCACTGCATG, Reverse-CCACAAGC TTCTGCCTCAGCCC; XPF: Forward-CCCAGCTCCTTCCCTTTCCCCA, Reverse-ACAACTCCGCCGTTGCATGAGG). Sanger sequencing was performed to determine the sequence of mutant alleles; XPG-E791A: AGAG(E) → GGCC(A) and XPF-D687A (additional R deletion at 701): GAT(D) → GCA(A) and TCG (R) deletion.

For *ERCC1*-knockout HAP1 cell lines: Exon 4, which encodes part of the XPA binding domain, was disrupted with a hygromycin B resistance marker. Two gRNA sequences targeting exon 4 (TTGCGCACGAACTT-CAGTAC and AATTACGTCGCCAAATTCCC) were cloned into pX330 vector. Sequences upstream and downstream of exon 4 were amplified with the following primers: GCCAAGCCCTTATTCCGATCTACAC and GAAGGGCAGAAGCCATCAATAGGG for the left arm, GTGAGCTCTGCGGCGCCACC and GGAATACTAAGGGCTCA-GAGTACGGC for the right arm. These left arm and right arm sequences were then cloned into the targeting vector DT-A-pA/loxP/PGK-Hygro-pA/loxP (Laboratory for Animal Resources and Genetic Engineering, Center for Developmental Biology, RIKEN Kobe, gift from Professor S. Takeda) flanking the hygromycin resistance gene. gRNA and targeting vectors were transfected into HAP1 cells (C859, Horizon) using Xfect Transfection Reagent (Takara). Integration of the hygromycin resistance marker in exon 4 was also confirmed by PCR with the following primers ATCTTTGTAGAAACCATCGGCGCAGCTATT (anneals to hygromycin resistance gene) GGGAGTTGAGAGGTCTCAGTCTCTTC (anneals to ERCC1 gene sequence downstream of the right arm).

### Trabectedin or UV irradiation for alkaline COMET chip assay

Cells were enriched at the G1 phase by incubating in growth medium supplemented with 1 μM (U2OS) or 2 μM (HAP1) palbociclib (hereafter referred to as the "working medium") for 24 h prior to exposure to trabectedin or UV irradiation. XP2YO, and XP2OS were asynchronous prior to exposure to trabectedin or UV irradiation. Following this, cells were embedded in a 30 μm COMET chip (catalog no. 4250-096-01, Trevigen) and further incubated for 30 min at 37 °C in the working medium. The medium was removed, and cells were incubated in the working medium supplemented with 50 nM trabectedin for 2 h in the presence and absence of repair synthesis inhibitors (HAP1: 0.5 mM HU, 5 μM AraC, U2OS: 1 mM HU, 10 μM AraC, XP2YO and XP2OS: 4 mM HU, 40 μM AraC). After trabectedin treatment, cells were incubated in the working medium either with or without the repair synthesis inhibitors for varying repair periods. In the case of UV treatment, after removal of the medium, cells were subjected to irradiation using 5 J/m² of UV-C (254 nm UV light). This was followed by incubation at 37 °C in the working medium, with or without the repair synthesis inhibitors, for different repair times. DNA strand breaks were examined utilizing the alkaline COMET chip assay.

### Alkaline COMET chip assay

The high-throughput variant alkaline COMET chip assay was performed as previously[23,47], with modification in the unwinding and electrophoresis steps.

*Embedding cells in COMET chip and cell lysis:* a 30 μm COMET chip (catalog no. 4250-096-01, Trevigen) was equilibrated in 100 mL tissue culture grade 1X PBS for 30 min at room temperature and placed into the 96-well COMET chip System (catalog no. 4260-096-CS, Trevigen). Single cell suspension was prepared in 6 mL working medium at

$1.0 \times 10^5$ cells/mL. Single cell suspension was aliquoted 100 μL per well. The lid-covered COMET chip System was placed in the tissue culture incubator for 10 min. Gently rocked E-W and N-S. Sit in the incubator another 10 min and repeat (total 30 min of incubation). The working medium was aspirated carefully not to remove cells. Cells were treated with either trabectedin or UV as described above. COMET chip with treated cells was then overlaid with 6 mL of 1% low melting agarose (catalog no. 4250-500-02, Trevigen), followed by cell lysis with 50 mL lysis solution (catalog no. 4250-500-01, Trevigen) for overnight at 4 °C.

*Unwinding of DNA and electrophoresis:* Unwinding of DNA was carried out for 30 min twice (HAP1, U2OS); 15 min twice (XP2YO, XP2OS) in 250 mL alkaline solution (200 mM NaOH, 1 mM EDTA, 0.1% Triton X-100). Electrophoresis was carried out for 50 min, 1 V/cm at 4 °C (HAP1, U2OS); 30 min, 1 V/cm at 4 °C (XP2YO, XP2OS) in 700 mL alkaline solution.

*Staining, imaging, and analysis:* After electrophoresis, COMET chip was neutralized for 15 min twice at 4 °C in 100 mL 0.4 M Tris pH 7.4 and equilibrated for 30 min at 4 °C in 100 mL 20 mM Tris pH 7.4, followed by staining in 50 mL 0.2 X SYBR Gold (catalog no. S11494, Invitrogen) at room temperature for 2 h. Stained COMET chip was destained at room temperature in 100 mL 20 mM Tris pH 7.4 up to 1 h.

Comets were imaged with 4X magnification on a fluorescence microscope (BX53, Olympus). % DNA in tail was quantified with Comet analysis software (catalog no. 4260-000-CS, Trevigen).

### Clonogenic survival assay

HAP1, U2OS, XP2YO, XP2OS, and XP3BR cells were cultured in growth media (refer to the Cell culture section). 1500 cells were seeded in triplicated 6 cm dishes a day before trabectedin or illudin S treatment. Cells were treated for 2 h with growth media containing either trabectedin or illudin S in varied concentration. Following treatment, media were changed to fresh growth media and cells were grown for 7–8 days. Cells were fixed with 4% paraformaldehyde for 15 min and stained with 1% methylene blue for 2 h. After washing with water, colonies (defined as ≥ 25 cells) were counted. The survival rate was normalized to the number of colonies of non-treated cells.

### Cell lysis and Western blotting

Cells were rinsed with ice-cold PBS and lysed in M-PER buffer (Thermo Fisher Scientific) containing Halt protease and phosphatase inhibitor cocktail (1X) (Thermo Fisher Scientific). The concentration of protein was determined using a Bio-Rad DC Protein Assay Kit (Bio-Rad). Samples were prepared by adding LDS sample buffer containing 2.5% of 2-mercaptoethanol (4X) (Invitrogen), followed by boiling at 95 °C for 5 min. Samples containing 25 μg of protein were resolved on 8–16% Tris-Glycine or 4–12% Bis-Tris gels (Thermo Fisher Scientific) at 150 V for 45 min, transferred onto Amersham Hybond 0.2 mm PVDF membrane (GE Healthcare) at 250 mA for 70 min with mini-protein tetra system (Bio-Rad). Transferred membranes were blocked with 5% skim milk in Tris-buffered saline (20 mM Tris Base, 137 mM NaCl, pH 7.6) containing 0.1% Tween 20 (TBS-T) for 1 h at room temperature (RT) and incubated with the following antibodies: α-β-actin (mouse, Invitrogen, catalog no. MA5-15739, 1:10,000), α-XPG (rabbit, Bethyl, A301-484A, 1:500), α -XPF for Supplementary Fig. 2a; left panel (rabbit, Abcam, ab76948 1:2,000) or for Supplementary Fig. 2a; right panel (mouse, 1:200 (Santa Cruz, sc-136153,), or α-ERCC1 (mouse, Santa Cruz, sc-17809, 1:300) were added to TBST and incubated overnight at 4 °C.

### Analysis of survival and COMET assay data

Quantitative values were expressed as mean ± SEM or mean ± SD. The exact sample size for each experiment is described in figure legends. Statistical significance of the survival and COMET assay data were analyzed by performing the ordinary two-way ANOVA or two-tailed paired t-tests. Differences between groups were considered significant

when $P < 0.05$. These statistical analyses were performed using GraphPad Prism (GraphPad Software) or Microsoft Excel.

## GLOE-seq library preparation

For the GLOE-seq positive control, i.e., checking the method's ability to identify DNA breaks introduced at known genomic locations (Supplementary Fig. 4a, b), genomic DNA was extracted using the Monarch genomic DNA purification kit (T3010) from untreated U2OS wildtype cells after harvesting (5 min incubation with 0.25 % trypsin). Existing breaks were blocked by incubating 5 µg DNA first with 10 units of T4 PNK per 4 ng DNA and 1x NEBuffer 2 for 30 min at 37 °C, and then with 250 µM ddNTP mix, 2 units of Therminator IX DNA polymerase and 1x ThermoPol buffer for 10 min at 60 °C. DNA was purified with ProNex size-purification system (NG2001, Promega) with 8:5 beads to sample ratio. DNA was incubated with 2.5 units of Nb.BsrDI and 1x CutSmart buffer for 90 min at 65 °C and then with 5 units antarctic phosphatase and 1x antarctic phosphatase buffer for 30 min at 37 °C, and purified as described above. DNA library for Illumina sequencing was prepared according to GLOE-Seq protocol 'Steps y23 – 28: Denaturation and ligation of 3′-OH termini (yeast)' onwards[29].

In the rest of GLOE-seq experiments, cells were grown up to 60–80% confluency and incubated for 24 h with 1 mM palbociclib isethionate for G1 enrichment. Cells were then treated with 0.1% DMSO, 20 nM or 50 nM trabectedin in combination with 1 mM palbociclib isethionate for 2 h. Cells were washed with PBS and underwent a 2-h recovery in cell culture media without trabectedin in the presence of 1 mM palbociclib isethionate. Nuclei isolation and DNA library preparation for sequencing followed the GLOE-seq protocol for mammalian cells[29] with the following modifications. *Isolation of genomic DNA from mammalian cell culture:* Agarose plugs were incubated with 6 mL of proteinase K solution and shaken at 170 rpm. *Fragmentation and capture of biotinylated single-stranded DNA:* DNA was sonicated with an average fragment length of 300 nucleotides with Qsonica for 5 min at 4 °C in cycles of 15 s ON and 5 s OFF with 20% amplitude. DNA was purified twice with AMPure beads with elution in 50 µL dH2O. Streptavidin MyOne C1 dynabeads were resuspended after washing in 50 µL bind and wash buffer (2x) for a 1:1 beads to sample ratio. *Second strand synthesis, end polishing and ligation of the distal adaptor:* Reagents for second strand synthesis, end polishing and distal adaptor ligation were combined on ice and mixed by pipetting. For the distal adaptor either 3792-UMI or 3792-UMI.v2 (different indexes) were annealed with 3791. DNA was purified with AMPure beads and eluted in 30 µL of dH2O. *qPCR:* Before the final library amplification described in[29], qPCR with reaction conditions identical to the final PCR used for library amplification was done for quality control. Samples (1 µL) were diluted in deionized water (9.4 µL) and combined with 1x Q5 buffer, 0.2 mM dNTPs, 0.1 µM i5 illumina indexing primer, 0.1 µM P7-short primer, 1x EvaGreen and 0.1 units of Q5 high-fidelity DNA polymerase. qPCR protocol included 120 s initial denaturation at 95 °C, 40 cycles of 15 s denaturation at 95 °C, 30 s annealing at 60 °C and 20 s extending at 72 °C, as well as 95 s final extension at 72 °C. *Library amplification:* The library was scaled up from 20 µL to 50 µL using entire samples. A customized P7-short primer compatible with the UMI-containing distal adaptors was used. Samples were purified with AMPure beads with a 1:1 bead-to-sample ratio. The DNA libraries were sequenced on an Illumina NovaSeq 6000 with a single-read protocol and the read length of 100 bp (R1); additionally, 15 cycles were run on R2 to read UMI and custom indexes introduced by the oligos 3792-UMI or 3792-UMI.v2. Supplementary Data 1 summarizes the reagents, enzymes, kits, oligonucleotides, and other materials used for GLOE-seq experiments.

## GLOE-seq data analysis

**Sequencing read processing.** After demultiplexing of sequencing data, each sample was represented by two fastq.gz files, with the first file containing 101-nucleotide-long genomic reads (R1) and the second storing the respective 10-nucleotide long UMIs (R2). The quality of the R1 and R2 data was checked using FastQC/0.11.9[48]. Low-quality and adaptor-containing reads were removed via trimmomatic/0.38[49], using the following parameters: for R1, SE ILLUMINACLIP:Trimmomatic-0.39/adapters/TruSeq3-SE.fa:2:30:10 LEADING:3 TRAILING:3 SLIDINGWINDOW:4:15 MINLEN:101; for R2, SLIDINGWINDOW:4:15 MINLEN:10. Using a custom Python/3.7.4 script employing the module Biopython/1.79, we merged the R1 and R2 records that passed trimming, incorporating the 10-nucleotide-long R2 sequences in the name of respective R1 reads. These reads were mapped to human reference genome GRCh38 via bowtie2/2.3.5.1, using the pre-built bowtie2[50] index from https://genome-idx.s3.amazonaws.com/bt/GRCh38_noalt_as.zip and default settings. Read duplicates were removed by the tool dedup of umi_tools/1.1.2 toolkit[51], grouping reads with the same R2 sequence stored in the read name (*method=unique*). samtools/1.12[52] were employed to remove unmapped reads, sort, index and generate statistics of bam files. bedtools2/2.29.2[53] was used to covert bam files to bed files. Each read represented one unit of DNA-break signal, which we positioned at the nucleotide located immediately upstream of the 5′ end of the read. Since a GLOE-Seq read is the reverse complement of the DNA fragment captured in the protocol[29], the strand of the nucleotide bearing the signal was changed to the opposite to the one on which the respective read was mapped. In this way signal positioning, the original DNA break is on the 5′ side of the nucleotide bearing the signal. Using AWK and a custom Python/3.7.4 script with the modules numpy/1.21.5 and pandas/0.25.1, we implemented the described DNA-break-signal positioning, which resulted in the deposited sample-specific tsv-files. In these tsv-files, each line reflects one DNA break revealed by one mapped read. The tsv-files contain the following columns: 1) the chromosome, 2) the 0-based coordinate and 3) the strand of the nucleotide bearing the signal (this nucleotide is immediately downstream of the break), 4) MAPQ score of the read. Supplementary Fig. 8 presents the evolution of read counts throughout major processing steps (read quality filtering, mapping, deduplication). The raw sequencing data and the tsv-files with called DNA breaks have been deposited in the NCBI Gene Expression Omnibus (GEO) under accession code GSE245883.

**Software for mapped DNA break data analysis.** The downstream analysis of DNA break data and their visualization were performed via custom Python/3.7.4 scripts and Jupyter notebooks employing the modules numpy/1.19.2, scipy/1.6.3, pandas/1.1.3, biopython/1.79, logomaker/0.8, matplotlib/3.4.2 and seaborn/0.11.1 in Python/3.8.5 environment. Besides, bedtools2/2.29.2[53] was used to extract the sequence context of DNA breaks from the reference genome (GRCh38). The custom code for genome-scale data analysis is available at https://gitlab.ethz.ch/eth_toxlab/trabi-seq.

**External datasets.** The following publicly available datasets were used in the analysis. Human reference genome, GRCh38 (pre-built bowtie2 index). Transcript coordinates: GENCODE/V41/knownGene, retrieved from UCSC Table Browser. Canonical transcripts of genes: GENCODE/V41/knownCanonical, retrieved from UCSC Table Browser. Gene expression: DepMap Public 22Q2 https://depmap.org/portal/download/all/?releasename=DepMap+Public+22Q2&filename=CCLE_expression_full.csv), the cell-line accession numbers ACH-000364 (U2OS WT) and ACH-002475 (HAP1 WT). Protein-coding genes: GENCODE/V41/knownToNextProt, retrieved from UCSC Table Browser. Coordinates of centromeres and gaps: retrieved from UCSC Table Browser for GRCh38. Chromatin accessibility and histone modification: from NCBI's GEO under accession number GSE87831[54], specifically, GSE87831_DNase-Seq.r1.peaks.bed.gz, GSE87831_DNase-Seq.r2.peaks.bed.gz, GSE87831_H3K4me3.peaks.bed.gz (referred to as H3K4me3 [1] in Fig. 3c); from GEO accession number GSE44672[55], specifically, GSM1356566_U2OS_H3K4me3.txt.gz (referred to as

H3K4me3 [2] in Fig. 3c), GSM1356565_U2OS_H3K4me1.txt.gz, GSM1356567_U2OS_H3K27ac.txt.gz; genomic coordinates were converted from hg19 to hg38 via https://genome.ucsc.edu/cgi-bin/hgLiftOver. Oncogenes: COSMIC Cancer Gene Census [https://cancer.sanger.ac.uk/census] (downloaded on 15.05.2023).

**Gene boundaries, DNA break count normalization and further figure details.** In our analysis, genes were represented by canonical transcripts (GENCODE/V41/knownCanonical), whose boundaries were considered as transcription start site (TSS) and transcription end site (TES). When operating with whole-gene features, we summed the signals mapped between TSS and TES. Genome-scale data required normalization of mapped signals since sequencing depth varied across samples (Supplementary Fig. 8). Therefore, for each sample, we calculated a normalization factor that reflects the level of endogenous DNA breaks in unexpressed genes. Specifically, the normalization factor was $M(s) = \text{mean}_{across\,f}\{\frac{N(s,f)}{L(f)}\}_f$, where features $f$ here are the transcribed (antisense) strands of not expressed genes, $N(s,f)$ is the number of mapped DNA breaks per a concrete feature $f$ in the sample $s$, and $L(f)$ is the feature's length measured in kilobases. DNA break count data $C(s,f)$ were related to these sample-specific normalization factors via the formula $C(s,f) = \frac{N(s,f)}{L(f)\cdot M(s)}$ [arb. unit], where a feature $f$ is a chromosome bin with both strands considered together (Fig. 3b) or separately (Fig. 5), a gene's transcribed strand or a gene's non-transcribed strand (Fig. 4a–d, Supplementary Fig. 5a–f), 5-kilobase region downstream or upstream of the TSS with both strands considered separately (Fig. 6g–h) or together (Supplementary Fig. 7). In Supplementary Fig. 4c, DNA break counts per bin were further related to the median value in a sample (fold change with respect to the median, $C(s,f)/\text{median}_{across\,f}\{C(s,f)\}_f$). In Fig. 4e–h and Supplementary Fig. 5g, we presented $\text{mean}_{across\,f}\{C(s,f)\}_f$.

For profiles of mean DNA break counts (Fig. 6a–f, Supplementary Fig. 6a–f), the following data were presented: $\text{mean}_{across\,g \in G}\{C(s,g,t,b)\}_g$ with $C(s,g,t,b) = \frac{N(s,g,t,b)}{L_g(b)\cdot M(s)}$, where $N(s,g,t,b)$ is the number of mapped DNA breaks per bin $b$ in sample $s$, gene $g$ and strand $t$, $L_g(b)$ is the bin's length measured in kilobases, $M(s)$ is the normalization factor, and $G$ is the set of top 30% expressed protein-coding genes in unexposed U2OS WT or HAP1 WT. When a bin size was absolute (provided as a base number), bins had the same size across all considered genes, $L_g(b) = L(b)$. When a bin size was relative (provided as a percentage, specifically, in the region between TSS and TES), each bin was the indicated fraction ($\alpha$) of the respective gene length, therefore, the bin size $L_g(b) = \alpha \cdot L(g)$ measured in bases varied across genes (the average value is indicated in parentheses in respective figure panels).

For Fig. 4, Supplementary Fig. 5a–d and Fig. 6g–h, in U2OS, we considered 16,740 protein-coding genes, including not expressed genes: 1989, ≤25% gene expression tier: 3689, ≤50%: 3689, ≤70%: 2948, ≤80%: 1475, ≤90%: 1475, ≤95%: 737, ≤100%: 738. Number of genes beyond the maximal Y-axis value: in Fig. 4a, 16 (TS, transcribed strand), 0 (NTS, non-transcribed strand); in Fig. 4b, 0 (TS), 1 (NTS); in Fig. 4c, 34 (TS), 0 (NTS); in Fig. 4d and Supplementary Fig. 5a,c,d, 0 (TS), 0 (NTS); in Supplementary Fig. 5b, 0 (TS), 2 (NTS). For Supplementary Fig. 5e–f, in HAP1, we considered 16,740 protein-coding genes, including not expressed genes: 3,428, ≤ 25% gene expression tier: 3334, ≤50%: 3324, ≤70%: 2660, ≤80%: 1331, ≤90%: 1331, ≤95%: 666, ≤100%: 666. Number of genes beyond the maximal Y-axis value: in Supplementary Fig. 5e, 19 (TS), 26 (NTS); in Supplementary Fig. 5f, 10 (TS), 3 (NTS).

For Fig. 6e–f, gene numbers per gene-length group: >50 Kb, 1070; ≤50 Kb, 1145; ≤22, 1072; ≤10 Kb, 1138. The strand-specific G content is calculated for the whole gene set (regardless of the gene length) using the reference genome and shows the absence of drastic sequence composition changes that would explain low DNA break counts in promoters.

For Fig. 6g–h, the plots are built analogously to Fig. 4a–d (lower panels), however, instead of the whole gene regions, the indicated 5-Kb regions are considered, which explains higher values for the transcribed strand as compared to Fig. 4a,c due to non-uniform break distribution throughout the gene body (Fig. 6a,c). Number of genes beyond the maximal Y-axis value: in Fig. 6g, 118 (TS), 27 (NTS); in Fig. 6h, 330 (TS), 84 (NTS).

## Reporting summary

Further information on research design is available in the Nature Portfolio Reporting Summary linked to this article.

## Data availability

The raw sequencing data and processed sequencing data (tsv-files with called DNA breaks) generated in this study have been deposited in the NCBI Gene Expression Omnibus (GEO) under GEO series accession code GSE245883. Source data are provided with this paper and have been deposited in Zenodo [https://doi.org/10.5281/zenodo.10477974]. Source data are provided with this paper.

## Code availability

The code for genome-scale data analysis required to generate respective figures is available at https://gitlab.ethz.ch/eth_toxlab/trabi-seq.

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

## Acknowledgements

Genome-scale data produced and analyzed in this paper were generated using the resources of the ETH Zürich Genetic Diversity Centre (GDC), the Functional Genomics Center Zürich (FGCZ), and the ETH Zürich Euler cluster. The authors thank members of Laboratory of Toxicology at ETH Zürich and the SNF Sinergia consortium "Hijacking Transcription-Coupled DNA Repair for Cancer Therapy" for productive discussion of the work. The authors are grateful to Martijn Luijsterburg (University of Leiden), and Hannes Lans, Jurgen Marteijn and Wim Vermeulen (Erasmus MC) for U2OS cell lines, Jiyoung Park (IBS-CGI) for help with establishing COMET chip assay, Navnit Kaur Singh, Laura Slappendel, Jasmina Büchel and Sabine Diedrich for help with experiments, and Sharon Cantor (UMass Worcester) for a critical reading of the manuscript. This work was supported by the Korean Institute of Basic Science (IBS-R022-A1 to ODS, IBS-R022-A2 to DI), the Swiss National Science Foundation (Sinergia grant CRSII5-186332 to SJS and ODS) and the Deutsche Forschungsgemeinschaft (DFG, German Research Foundation – Project-ID 393547839 – SFB 1361 to HDU).

## Author contributions

KS, VT, SJS, and ODS conceptualized research. KS, VM, and HY conducted COMET chip and survival assays. KS, HY, and DI generated cell lines. VT designed experiments for genome-scale DNA break mapping and analyzed the respective data. ED prepared GLOE-seq libraries. NJLP helped to implement the GLOE-seq protocol. NZ and HDU upgraded the GLOE-seq protocol and advised on its implementation. KS, VT, and ODS wrote the original manuscript draft, and it was reviewed by all the authors. ODS, SJS and HDU supervised research and acquired funding.

## Competing interests

The authors declare no competing interests.
