## [Peer Review File · Nature Communications]

Trabectedin derails transcription-coupled nucleotide excision repair to induce DNA breaks in highly transcribed genesREVIEWER COMMENTS

Reviewer #1 (Remarks to the Author):

Trabectedin is a unique anticancer agent that forms a covalent N2-dG adduct and induces cytotoxic DNA breaks in TC-NER proficient cells. Due to its stabilising effect on the DNA duplex, trabectedin-DNA adducts evade recognition by GG-NER. Consequently, the damage is exclusively repaired by TC-NER, with cytotoxicity solely dependent on TCR factors. While previous studies have yielded clinically relevant insights into trabectedin's properties, our understanding of the exact mechanisms driving its toxicity remains limited.

In this manuscript, the authors aim to elucidate detailed mechanistic insights into the repair of trabectedin-DNA adduct through the TCR pathway. The authors have introduced several state-of-the-art techniques and demonstrated that trabectedin's toxicity arises from a single-strand-break (SSB) repair intermediate, a consequence of inhibiting the 3' incision process of NER. Using high-throughput alkaline COMET-chip assay, the authors initially showed that trabectedin treatments induce abortive TC-NER, leading to the formation of persistent unrepaired SSB intermediates. With the COMET-chip system on their incision-deficient Δ NER cell strains, the authors have pointed out that the 5'-NER endonuclease, XPF/ERCC1, is crucial for the induction of SSBs following trabectedin treatment. The authors have also noted that XPG is indispensable for the SSB formation, but its 3'-endonuclease activity is unnecessary. Moreover, by using a next-generation-sequencing (NGS)-based approach (GLOE-seq), the authors have successfully mapped the 3'-OH of trabectedin-induced breaks genome-wide. Collectively, the authors have demonstrated that trabectedin induced SSB preferentially occurs in the transcribed strands of the highly expressed genes.

This study unquestionably offers new insights into the pharmacological action of trabectedin. Furthermore, a detailed understanding of how trabectedin interferes with the 3'-incision process by XPG may contribute to the further development of anticancer therapeutics targeting the TCR pathway. The experiments are well-designed and conducted appropriately. In particular, TRABI-seq (GLOE-seq) data shown in Figure 3 is impressive. Despite some weaknesses, such as the absence of molecular biological studies supporting their findings, the manuscript is high quality and is suitable for publication in Nature Communications with some amendments. The authors may consider to address either of the comments below for improving the manuscript before publication.

Comments:

Figure 1b: Δ CSB cells are compromised in the initiation of TCR, and in these cells, the recruitment of TFIIH to the damage-stalled RNAPII does not occur. Why are Δ XPA cells less sensitive to trabectedin than Δ CSB cells?

Figure 1b, 1d, 1e: It is unfortunate that the authors only used Δ CSB and Δ XPA cells as representative TCR-deficient cell lines for the assays. The authors might consider including Δ UVSSA cells to investigate whether trabectedin also interferes with the recruitment of TFIIH by UVSSA. I assume not, but if Δ UVSSA does not affect the trabectedin toxicity, that is also intriguing.

Figure 2: The experiments clearly demonstrate that the 5'-endonuclease activity of the XPF/ERCC1 is needed for the trabectedin toxicity, but 3'-endonuclease activity of XPG is dispensable. An intriguing question to answer is whether the helicase activities of TFIIH/XPB/XPD are necessary for the toxicity as a consequence of the induction of persistent unrepaired SSB intermediates.

Figure 2f: HAP1 experiments always give ambiguous output compared to those of U2OS. Consider to replace this panel with data using U2OS.

Figure 2: TCR is initiated by the ubiquitination of RNAPII-RPB1 stalled at DNA lesions. Either of the treatments with NEDD8, VCP/p97, or ubiquitination inhibitors may interfere with the recruitment of TCR factors followed by trabectedin treatments. The authors may set up experiments to determine whether the inhibition of RNAPII ubiquitination can evade the trabectedin cytotoxicity.

Figure 3b, 3d-3g: TRABI-seq is impressive. The authors may consider to include pictures for

representative genes as well as for the meta gene profiles.

Figure 3i, 3k: Δ CSB completely lacks dose dependency for trabectedin induced SSB formation, while Δ XPA remains some. Is this due to an incision without damage verification by XPA?

Reviewer #2 (Remarks to the Author):

Trabectedin(ET743), forms adducts with DNA at the N2-position of dG. These lesions have been shown to not be recognized by global genome nucleotide excision repair (GG-NER), but instead appear to induce lethality through blocking transcriptional coupled-NER (TC-NER). This remarkable property of this compound has allowed its use in the clinic as an antitumor agent in the treatment of sarcoma and ovarian cancer. This present strong study by this outstanding team has investigated the mechanism of how ET743 inhibits-TC-NER. After showing confirmatory survival data ruling out GG-NER and showing that lethality depends on an active TC-NER, the authors show the appearance of DNA strand breaks using alkaline COMET assay over time. They have adapted a AraC-HU assay to prevent DNA polymerase incorporation and holding open the DNA repair-initiated event. The authors then use KO and KI cell lines deficient in nuclease activity of either XPF-ERCC1 or XPG. They further show that both nucleases are necessary to initiate nicks at sites of ET743. However, the structure of the ET74-lesion appears to block the XPG nuclease from completing the excision. The XPA is an important negative control. The cartoons in Figures 1 and 2 are very helpful to the reader. The authors then adapted GLOE-seq in U2OS cells to map the 3'-OH sites of the strand breaks across the genome after treatment with ET743, which they have given the name, TRABI-seq. This innovative and robust assay allowed the authors to demonstrate that: 1) strand breaks accumulate at the beginning of genes on the transcribed strand, 2) strand breaks were proportional to the gene expression levels, and 3) on the opposite strand in the other direction; this divergent transcription pattern has been noted previously. Finally the authors show a sequence preference for XPF-ERCC1 incisions. Overall, the experiments are well executed, and the methodology detailed. The conclusions drawn follow from the data presented. The manuscript is well-written, and an important contribution to the field. The authors are asked to consider the following points, which could enhance this already strong study.

1. Figure 1. Showing the structure of ET743 is useful, but having a better understanding of how the N2-dG adduct is situated in the DNA would enhance this figure.
2. While the supplemental is robust and necessary, the quantitative data panels d-j, in Supplemental figure 1, should be combined with the representative COMETs shown in Figure 1 and could replace d which do not give you the best feel of the overall data trends.
3. Why are the results in Figure 2 more robust in the U2OS cell lines as compared to the HAP1 cell line?
4. Not clear that all of the data in Figure 3 is necessary to shown in the main figures, for example panels d-g.

Reviewer #3 (Remarks to the Author):

In this manuscript, the authors determined that abortive TC-NER of trabectedin-DNA adducts forms persistent single-strand breaks (SSBs). XPF-ERCC1 endonuclease is necessary for trabectedin-induced SSBs and the trabectedin-DNA adduct inhibits XPG catalytic activity. The authors mapped the persistent ERCC1-XPF-dependent SSBs upstream of the trabectedin-DNA adduct by the upgraded GLOE-Seq method. They found that trabectedin-induced SSBs primarily occur near TSS sites and outside gene bodies, connecting TC-NER to divergent transcription from promoters.

Here are some concerns:

1. In page 10, the precision (true positive) of the upgraded GLOE-Seq in this paper was 82-84%, while in ref. 28 (Mol Cell 78, 975-985 e7 (2020)) the precision was above 90%. Besides, what's the false positive of the upgraded GLOE-Seq method?
2. The authors mapped trabectedin-induced SSBs by GLOE-Seq method but the information such as the total reads, mapped reads and the mapping efficiency for each sequencing libraries was not provided.
3. It's interesting that trabectedin-induced SSBs primarily occur near TSS sites and correlation with transcriptionally active regions of chromatin. What may contribute to this biased distribution? What's the relation of trabectedin-induced SSBs with transcriptionally inactive regions of chromatin?
4. In page 10 line 222-225, the authors claimed "abundant trabectedin-induced DNA breaks in the TC-NER-proficient cells" and "no trabectedin-induced DNA breaks in the TC-NER-deficient cells". The numbers of SSBs in different cells should be shown. The conclusion of "no trabectedin-induced DNA breaks in the TC-NER-deficient cells" is lacking in rigor.

In response to Reviewer #1:

Trabectedin is a unique anticancer agent that forms a covalent N2-dG adduct and induces cytotoxic DNA breaks in TC-NER proficient cells. Due to its stabilising effect on the DNA duplex, trabectedin-DNA adducts evade recognition by GG-NER. Consequently, the damage is exclusively repaired by TC-NER, with cytotoxicity solely dependent on TCR factors. While previous studies have yielded clinically relevant insights into trabectedin's properties, our understanding of the exact mechanisms driving its toxicity remains limited.

In this manuscript, the authors aim to elucidate detailed mechanistic insights into the repair of trabectedin-DNA adduct through the TCR pathway. The authors have introduced several state-of-the-art techniques and demonstrated that trabectedin's toxicity arises from a single-strand-break (SSB) repair intermediate, a consequence of inhibiting the 3' incision process of NER. Using high-throughput alkaline COMET-chip assay, the authors initially showed that trabectedin treatments induce abortive TC-NER, leading to the formation of persistent unrepaired SSB intermediates. With the COMET-chip system on their incision-deficient Δ NER cell strains, the authors have pointed out that the 5'-NER endonuclease, XPF/ERCC1, is crucial for the induction of SSBs following trabectedin treatment. The authors have also noted that XPG is indispensable for the SSB formation, but its 3'-endonuclease activity is unnecessary. Moreover, by using a next-generation-sequencing (NGS)-based approach (GLOE-seq), the authors have successfully mapped the 3'-OH of trabectedin-induced breaks genome-wide. Collectively, the authors have demonstrated that trabectedin induced SSB preferentially occurs in the transcribed strands of the highly expressed genes.

This study unquestionably offers new insights into the pharmacological action of trabectedin. Furthermore, a detailed understanding of how trabectedin interferes with the 3'-incision process by XPG may contribute to the further development of anticancer therapeutics targeting the TCR pathway. The experiments are well-designed and conducted appropriately. In particular, TRABI-seq (GLOE-seq) data shown in Figure 3 is impressive. Despite some weaknesses, such as the absence of molecular biological studies supporting their findings, the manuscript is high quality and is suitable for publication in Nature Communications with some amendments. The authors may consider to address either of the comments below for improving the manuscript before publication.

Thank you very much for the positive assessment of our study.

Comments:

1. Figure 1b: Δ CSB cells are compromised in the initiation of TCR, and in these cells, the recruitment of TFIIH to the damage-stalled RNAPII does not occur. Why are Δ XPA cells less sensitive to trabectedin than Δ CSB cells?

The reviewer raises an interesting point. The answer to this question warrants further investigation that we will conduct as part of a future manuscript. Our current hypothesis is that in the absence of CSB, a fraction of RNA Pol II stalled at lesions is not degraded, contributing to trabectedin toxicity. This may partially compensate for the resistance resulting from a lack of break formation in the absence of CSB and TC-NER. We consider this type of analysis and discussion, while important, to be outside of the scope of the current manuscript.

2. Figure 1b, 1d, 1e: It is unfortunate that the authors only used Δ CSB and Δ XPA cells as representative TCR-deficient cell lines for the assays. The authors might consider including Δ UVSSA cells to investigate whether trabectedin also interferes with the recruitment of TFIIH by UVSSA. I assume not, but if Δ UVSSA does not affect the trabectedin toxicity, that is also intriguing.

We have preliminary data from clonogenic assay for UVSSA cells that are similar to the results of other TC-NER deficient cells (see figure below). We did not include them in the current manuscript as there was no dramatic difference to CSB cells. Generating publication quality data would require an estimated 1-2 months of experiments. We are planning to further the nuances of the role of TC-NER in trabectedin toxicity in a future manuscript also relating to previous point raised by the reviewer.

3. Figure 2: The experiments clearly demonstrate that the 5'-endonuclease activity of the XPF/ERCC1 is needed for the trabectedin toxicity, but 3'-endonuclease activity of XPG is dispensable. An intriguing question to answer is whether the helicase activities of TFIIH/XPB/XPD are necessary for the toxicity as a consequence of the induction of persistent unrepaired SSB intermediates.

This is an interesting point. Considering that both XPB and XPD ATPase activities are required for NER, we predict that both activities will be needed for trabectedin toxicity. This rather detailed mechanistic point might be addressed in our ongoing biochemical studies of the activity of NER proteins on trabectedin-DNA lesions. We consider such a detailed analysis to be out of scope for this manuscript.

4. Figure 2f: HAP1 experiments always give ambiguous output compared to those of U2OS. Consider to replace this panel with data using U2OS.

The reviewer is correct that the difference in break formation as measured by COMET chip (Fig 1, 2) and GLOE-Seq (Fig 3, 4, S5, S6) is quantitatively smaller in WT vs mutant in HAP1 vs U2OS cells. We note however, that the observations in the different phenotypes are 100% consistent between HAP1 and U2OS cells. In Fig 2, we relied on HAP1 cells as their haploid genome allowed us to readily generate an active site mutation at the genomic loci of XPF and XPG. The results in Figure 2 are in our opinion very clear and not ambiguous, although quantitatively different from similar experiments in U2OS cells. Repeating the experiments in U2OS would therefore yield no dramatic new insights but would require us to invest months of work to generate new sets of cell lines with point mutations in XPF and XPG.

However, we do now address the quantitative difference between the data from U2OS and HAP1 cells is now discussed in conjunction with Figures 1j and k.

5. Figure 2: TCR is initiated by the ubiquitination of RNAPII-RPB1 stalled at DNA lesions. Either of the treatments with NEDD8, VCP/p97, or ubiquitination inhibitors may interfere with the recruitment of TCR factors followed by trabectedin treatments. The authors may set up experiments to determine whether the inhibition of RNAPII ubiquitination can evade the trabectedin cytotoxicity.

Excellent point for future studies to get a better understanding on the interaction of Trabectedin-DNA adducts with RNA polymerases. Fundamentally, these experiments will not change the conclusion that the break formation/toxicity of trabectedin-DNA adducts is TC-NER dependent and are better suited for a future manuscript (see also our reply to points 1 and 2 of reviewer 1).

6. Figure 3b, 3d-3g: TRABI-seq is impressive. The authors may consider to include pictures for

representative genes as well as for the meta gene profiles.

Thank you for this excellent suggestion. *We have added a new panel **Figure 4a** with DNA-break mapping along five most damaged genes following trabectedin exposure and described it in the text.*

7. Figure 3i, 3k: Δ CSB completely lacks dose dependency for trabectedin induced SSB formation, while Δ XPA remains some. Is this due to an incision without damage verification by XPA?

*We do not currently understand the reason for this observation. In canonical NER, XPA is essential for complex assembly and dual incision, so we favor the idea that NER-independent incisions may occur in rare cases or at a subset of trabectedin lesions in a transcription-dependent manner. Future investigations will be needed to investigate this observation that are beyond the scope of the current study. We added a sentence addressing this point in our description of **Figures 3h and i**.*

Reviewer #2 (Remarks to the Author):

Trabectedin (ET743), forms adducts with DNA at the N2-position of dG. These lesions have been shown to not be recognized by global genome nucleotide excision repair (GG-NER), but instead appear to induce lethality through blocking transcriptional coupled-NER (TC-NER). This remarkable property of this compound has allowed its use in the clinic as an antitumor agent in the treatment of sarcoma and ovarian cancer. This present strong study by this outstanding team has investigated the mechanism of how ET743 inhibits-TC-NER. After showing confirmatory survival data ruling out GG-NER and showing that lethality depends on an active TC-NER, the authors show the appearance of DNA strand breaks using alkaline COMET assay over time. They have adapted a AraC-HU assay to prevent DNA polymerase incorporation and holding open the DNA repair-initiated event. The authors then use KO and KI cell lines deficient in nuclease activity of either XPF-ERCC1 or XPG. They further show that both nucleases are necessary to initiate nicks at sites of ET743. However, the structure of the ET74-lesion appears to block the XPG nuclease from completing the excision. The XPA is an important negative control. The cartoons in Figures 1 and 2 are very helpful to the reader. The authors then adapted GLOE-seq in U2OS cells to map the 3'-OH sites of the strand breaks across the genome after treatment with ET743, which they have given the name, TRABI-seq. This innovative and robust assay allowed the authors to demonstrate that: 1) strand breaks accumulate at the beginning of genes on the transcribed strand, 2) strand breaks were proportional to the gene expression levels, and 3) on the opposite strand in the other direction; this divergent transcription pattern has been noted previously. Finally the authors show a sequence preference for XPF-ERCC1 incisions. Overall, the experiments are well executed, and the methodology detailed. The conclusions drawn follow from the data presented. The manuscript is well-written, and an important contribution to the field. The authors are asked to consider the following points, which could enhance this already strong study.

We appreciate your supportive comments on our work.

1. Figure 1. Showing the structure of ET743 is useful, but having a better understanding of how the N2-dG adduct is situated in the DNA would enhance this figure.

*Thank you for this suggestion. We modified **Fig. 1a** to include a 3D model of the trabectedin-DNA lesion that shows its bulky, non-distorting nature.*

2. While the supplemental is robust and necessary, the quantitative data panels d-j, in Supplemental figure 1, should be combined with the representative COMETs shown in Figure 1 and could replaced which do not give you the best feel of the overall data trends.

*Thank you for this suggestion and appreciation of our data. As suggested, we have moved most of the quantitative raw COMET-chip data from **Supplementary Fig. 1** to the main **Fig 1**, introducing*

new panels d-i. We have kept the graphs as panels Fig. 1j-k as they show experimental variability and allow comparison across cell lines, illustrating the quantitative difference of the COMET data for HAP1 and U2OS cells.

3. Why are the results in Figure 2 more robust in the U2OS cell lines as compared to the HAP1 cell line?

We consistently and quantitatively see differences in signal intensities for the COMET chip assays as well as TRABI-Seq between U2OS and HAP1 cells. While the cause is not known, this may be due to differences in overall transcription levels. Since the differences are consistent throughout various experiments and genotypes, it does not impact any of our conclusions. *We discussed the quantitative difference between the data from U2OS and HAP1 cells in the manuscript in conjunction with Figures 1j and k (see also point 4 of reviewer 1).*

4. Not clear that all of the data in Figure 3 is necessary to shown in the main figures, for example panels d-g.

The reviewer is correct that Fig. 3d-g, containing three panels each, show a lot of data. We do, however, feel that it is advantageous to have the original and binned data representations together so that the reader can appreciate the variability of damage among genes and fully understand our analysis steps. The panels Fig. 3d-g present damage distributions in one biological replicate of cell-line–exposure combination. On the other hand, Fig. 3h-k show mean values from all biological replicates of all cell-line–exposure combinations. The panels in Fig. 3d-g thus explain where the values in Fig. 3h-k come from.

Reviewer #3 (Remarks to the Author):

In this manuscript, the authors determined that abortive TC-NER of trabectedin-DNA adducts forms persistent single-strand breaks (SSBs). XPF-ERCC1 endonuclease is necessary for trabectedin-induced SSBs and the trabectedin-DNA adduct inhibits XPG catalytic activity. The authors mapped the persistent ERCC1-XPF-dependent SSBs upstream of the trabectedin-DNA adduct by the upgraded GLOE-Seq method. They found that trabectedin-induced SSBs primarily occur near TSS sites and outside gene bodies, connecting TC-NER to divergent transcription from promoters.

Thank you for the concise summary of our work.

Here are some concerns:

1. In page 10, the precision (true positive) of the upgraded GLOE-Seq in this paper was 82-84%, while in ref. 28 (Mol Cell 78, 975-985 e7 (2020)) the precision was above 90%. Besides, what's the false positive of the upgraded GLOE-Seq method?

We estimated the precision, *i.e.*, the ratio between the number of true positive calls and the number of all calls, as the fraction of reads revealing the CATTGC pattern (the recognition site of the nickase Nb.BsrDI) out of all reads in a sample. The precision of the upgraded GLOE-Seq in this manuscript is 82-84%. In the original GLOE-Seq paper, the precision was around 60% (not 90% as indicated by the reviewer) – after digestion with Nb.BsrDI, more than 60% of total reads corresponded to Nb.BsrDI recognition sites (page 977 in Mol Cell 78, 975-985 e7 (2020)). Therefore, the false discovery rate (or 1 - precision), *i.e.*, the ratio between the number of *false* positive calls and the number of all calls, is 16-18% in the current manuscript and around 40% in the original publication of GLOE-Seq.

On the other hand, the sensitivity, which was assessed as the ratio between the number of identified Nb.BsrDI sites and the number of all Nb.BsrDI sites in the reference genome, is around 90% in the original method tested in the yeast genome and 87-88% in the upgraded method tested in the human genome.

Thus, observing better precision and comparable sensitivity, we concluded that the upgraded method performed on the human genome at least as well as the original GLOE-Seq protocol, which was validated in the same way in the yeast genome.

*We have introduced clarifications in the respective sentences in the discussion of **Figures S4a-b** of the manuscript.*

2. The authors mapped trabectedin-induced SSBs by GLOE-Seq method but the information such as the total reads, mapped reads and the mapping efficiency for each sequencing libraries was not provided.

Thank you for bringing this point to our attention. At the time of the manuscript submission to the journal, these data were available in the file *Read_counts_at_preprocessing_steps_all_samples_MS.csv* on this project's GitLab page https://gitlab.ethz.ch/eth_toxlab/trabi-seq. We fully agree with the reviewer that it would be useful to provide this information in the manuscript itself. *This data is currently shown in the new **Supplementary Figure 8***. For example, one can see there that the mapping efficiency is around 97% on average across the samples.

3. It's interesting that trabectedin-induced SSBs primarily occur near TSS sites and correlation with transcriptionally active regions of chromatin. What may contribute to this biased distribution? What's the relation of trabectedin-induced SSBs with transcriptionally inactive regions of chromatin?

We are not sure what the concern of the reviewer is. We show that trabectedin-induced breaks exclusively occur in the transcribed regions of the genome, in active genes or regions of divergent gene transcription. We do not observe any trabectedin-induced SSBs in G1 arrested cells in transcriptionally inactive regions of chromatin (Figure 3d,h). During replication, trabectedin lesions, like any other DNA adducts may stall replication and induce gaps, due separation of helicase and polymerase activities, but this is not a subject of investigation in the current manuscript.

4. In page 10 line 222-225, the authors claimed "abundant trabectedin-induced DNA breaks in the TC-NER-proficient cells" and "no trabectedin-induced DNA breaks in the TC-NER-deficient cells". The numbers of SSBs in different cells should be shown. The conclusion of "no trabectedin-induced DNA breaks in the TC-NER-deficient cells" is lacking in rigor.

Thank you for this excellent point. *We have added the requested quantification of the number of breaks by TRABI-Seq; there is a **new Supplementary Fig. 4d**, and we provided total DNA-break values for the conditions with and without trabectedin in **Fig. 3b***. Additionally, we conducted a statistical test comparing genome-wide DNA-break distributions between trabectedin and control exposures to provide further evidence that the overall increase in break counts is attributed to TC-NER of trabectedin-DNA adducts. *As a result, we generated a new **Supplementary Fig. 4e** and indicated the test p-values in **Fig 3b***.

REVIEWERS' COMMENTS

Reviewer #1 (Remarks to the Author):

While the authors' rebuttals are generally understandable, a weakness in the genetic studies lies in the absence of mechanistic insights. I still do not fully comprehend the survival difference between XPA-KO and CSB-KO cells as observed in Fig. 1b (***) represents statistical difference between mutants). To support the authors' conclusion regarding the trabectedin toxicity originated from the NER damage incision step and RNAPII-stalling, it would be beneficial to briefly address the following point in Discussion or somewhere with appropriate references for clarity before publication.

In response to my comments #1, #2, and #5, the authors explained that 'the absence of CSB results in a fraction of RNAPII stalled at lesions not being degraded, contributing to trabectedin toxicity,' 'UVSSA-KO cells exhibit results similar to other TC-NER deficient cells (insensitive like CSB-KO),' and 'the break formation/toxicity of trabectedin-DNA adducts is TC-NER dependent.'

I agree with the authors' acknowledgment that the UVSSA-KO data is not of publication quality. However, given the authors' conclusion attributing trabectedin toxicity to the abortive TC-NER, it is worthwhile to differentiate the working points of TC-NER factors concerning the degree of trabectedin insensitivity.

UVSSA-KO cells are TC-NER deficient, but damage-stalled RNAPII is degraded. Therefore, the compensation seen in CSB-KO cells, where the fraction of RNA Pol II stalled at lesions cannot offset the resistance, is lacking. Consequently, XPA-KO and UVSSA-KO should manifest similar insensitivity to trabectedin treatment. This assumption might not be true as CSB-KO, UVSSA-KO, and XPA-KO often exhibit similar damage responses in various endpoints such as survival/RRS after UV treatment. However, it is noteworthy that in some cases, CSB-KO displays a much stronger phenotype, as observed in Cockayne syndrome patients in humans than other TC-NER-deficient cases, such as UV-sensitive syndrome lacking UVSSA and XP-A. I assume that CSB-KO induces 'TFIIH-independent, direct abortive TC-NER' due to persistent RNAPII-stalling after trabectedin.

Reviewer #2 (Remarks to the Author):

The authors have responded well to the three reviews, and in some cases providing additional analyses as requested. The manuscript and study have been strengthened. The manuscript in its present form is suitable for publication.

Reviewer #3 (Remarks to the Author):

I have no further questions.

Reviewer #1 (Remarks to the Author):

While the authors' rebuttals are generally understandable, a weakness in the genetic studies lies in the absence of mechanistic insights. I still do not fully comprehend the survival difference between XPA-KO and CSB-KO cells as observed in Fig. 1b (***) represents statistical difference between mutants). To support the authors' conclusion regarding the trabectedin toxicity originated from the NER damage incision step and RNAPII-stalling, it would be beneficial to briefly address the following point in Discussion or somewhere with appropriate references for clarity before publication.

In response to my comments #1, #2, and #5, the authors explained that 'the absence of CSB results in a fraction of RNAPII stalled at lesions not being degraded, contributing to trabectedin toxicity,' 'UVSSA-KO cells exhibit results similar to other TC-NER deficient cells (insensitive like CSB-KO),' and 'the break formation/toxicity of trabectedin-DNA adducts is TC-NER dependent.'

I agree with the authors' acknowledgment that the UVSSA-KO data is not of publication quality. However, given the authors' conclusion attributing trabectedin toxicity to the abortive TC-NER, it is worthwhile to differentiate the working points of TC-NER factors concerning the degree of trabectedin insensitivity.

UVSSA-KO cells are TC-NER deficient, but damage-stalled RNAPII is degraded. Therefore, the compensation seen in CSB-KO cells, where the fraction of RNA Pol II stalled at lesions cannot offset the resistance, is lacking. Consequently, XPA-KO and UVSSA-KO should manifest similar insensitivity to trabectedin treatment. This assumption might not be true as CSB-KO, UVSSA-KO, and XPA-KO often exhibit similar damage responses in various endpoints such as survival/RRS after UV treatment. However, it is noteworthy that in some cases, CSB-KO displays a much stronger phenotype, as observed in Cockayne syndrome patients in humans than other TC-NER-deficient cases, such as UV-sensitive syndrome lacking UVSSA and XP-A. I assume that CSB-KO induces 'TFIIH-independent, direct abortive TC-NER' due to persistent RNAPII-stalling after trabectedin.

The reviewer is correct on insisting on this important point – that the phenotypes of cells is influenced by whether RNAPII is degraded in XPA-KO and UVSSA-KO cells, but not in CSB-KO. This may result in a different response and level of toxicity in XPA/UVSSA deficient cells versus CSB deficient cells, as we observe for XPA vs CSB in Fig 1b. This is indeed an issue that needs to be addressed and we are currently doing so in additional cell lines and will report the results in due course.

We have added a sentence to the discussion section to address this point:

“We note that we have not yet fully explored the nuances of how various TC-NER genes affect trabectedin-induced toxicity and lesion processing. For example, it is known that CSB is involved in RNA Polymerase II degradation and repair, while other TC-NER specific factors, such as UV-SSA specifically contribute to repair. It will be intriguing to explore in more detail how these properties influence cellular responses to trabectedin.”

We are looking forward to reporting a more detailed experimental investigation of this point in various cell lines in a future manuscript.

Reviewer #2 (Remarks to the Author):

The authors have responded well to the three reviews, and in some cases providing additional analyses as requested. The manuscript and study have been strengthened. The manuscript in it's present form is suitable for publication.

We thank reviewer 2 for the supportive comments.

Reviewer #3 (Remarks to the Author):

I have no further questions.

We thank reviewer 3 for the supportive comments.